# Developing Ni single-atom sites in carbon nitride for efficient photocatalytic $H_2O_2$ production

Xu Zhang[1,10], Hui Su[2,3,10], Peixin Cui[4], Yongyong Cao[5], Zhenyuan Teng[6], Qitao Zhang[7], Yang Wang[8], Yibo Feng[1], Ran Feng[1], Jixiang Hou[1], Xiyuan Zhou[1], Peijie Ma[1], Hanwen Hu[1], Kaiwen Wang[1], Cong Wang[1], Liyong Gan[8], Yunxuan Zhao[9], Qinghua Liu[2]✉, Tierui Zhang[9]✉ & Kun Zheng[1]✉

Photocatalytic two-electron oxygen reduction to produce high-value hydrogen peroxide ($H_2O_2$) is gaining popularity as a promising avenue of research. However, structural evolution mechanisms of catalytically active sites in the entire photosynthetic $H_2O_2$ system remains unclear and seriously hinders the development of highly-active and stable $H_2O_2$ photocatalysts. Herein, we report a high-loading Ni single-atom photocatalyst for efficient $H_2O_2$ synthesis in pure water, achieving an apparent quantum yield of 10.9% at 420 nm and a solar-to-chemical conversion efficiency of 0.82%. Importantly, using in situ synchrotron X-ray absorption spectroscopy and Raman spectroscopy we directly observe that initial Ni-$N_3$ sites dynamically transform into high-valent $O_1$-Ni-$N_2$ sites after $O_2$ adsorption and further evolve to form a key *OOH intermediate before finally forming HOO-Ni-$N_2$. Theoretical calculations and experiments further reveal that the evolution of the active sites structure reduces the formation energy barrier of *OOH and suppresses the O=O bond dissociation, leading to improved $H_2O_2$ production activity and selectivity.

As one of the top 100 chemicals in the world, hydrogen peroxide ($H_2O_2$) is a high-value green oxidant and an emerging clean liquid fuel[1,2]. Meanwhile, $H_2O_2$ is widely used in medical sterilization, printing, bleaching, waste water treatment and other fields, which is closely related to human life and social development[3–6]. At present, the traditional anthraquinone method for large-scale industrial production of $H_2O_2$ has serious shortcomings such as high energy consumption, intensive waste, and toxic by-products[7,8]. Photocatalytic $O_2$ reduction to $H_2O_2$ by solar-driven semiconductor catalysts using $H_2O$ and $O_2$ (2e- ORR, $O_2 + 2e^- + 2H^+ \rightarrow H_2O_2$) is a green, economical, and promising $H_2O_2$ production strategy, which has been attracting extensive attention[3,9–12]. Especially, photocatalytic efficient synthesis of $H_2O_2$ in pure water (without adding any sacrificial agent or buffer salt solution), which not only saves costs but also ensures the subsequent practical

[1]Beijing Key Laboratory of Microstructure and Properties of Solids, Faculty of Materials and Manufacturing, Beijing University of Technology, Beijing 100124, China. [2]National Synchrotron Radiation Laboratory, University of Science and Technology of China, Hefei 230029 Anhui, China. [3]College of Chemistry and Chemical Engineering, Hunan Normal University, Changsha 410081 Hunan, China. [4]Key Laboratory of Soil Environment and Pollution Remediation, Institute of Soil Science, Chinese Academy of Sciences, 210008 Nanjing, China. [5]College of Biological, Chemical Science and Engineering, Jiaxing University, Jiaxing 314001 Zhejiang, China. [6]School of Chemistry, Chemical Engineering and Biotechnology, Nanyang Technological University, Singapore 637459, Singapore. [7]International Collaborative Laboratory of 2D Materials for Optoelectronics Science and Technology of Ministry of Education, Institute of Microscale Optoelectronics, Shenzhen University, Shenzhen 518060, China. [8]College of Physics and Institute of Advanced Interdisciplinary Studies, Chongqing University, Chongqing 400044, China. [9]Key Laboratory of Photochemical Conversion and Optoelectronic Materials, Technical Institute of Physics and Chemistry, Chinese Academy of Sciences, Beijing, China. [10]These authors contributed equally: Xu Zhang, Hui Su. ✉e-mail: qhliu@ustc.edu.cn; tierui@mail.ipc.ac.cn; kunzheng@bjut.edu.cn

application of high-purity $H_2O_2$ (avoiding complicated and expensive distillation purification), is one of the goals pursued in this field[4,9,10,13–17]. Among various photocatalysts, the low-cost graphitic carbon nitride (g-$C_3N_4$) has shown great potential for $H_2O_2$ production due to the certain light-responsive ability, suitable energy band structure, and metal-free structure suppression of $H_2O_2$ surface decomposition[10,14,18–21]. However, the highest solar-to-chemical conversion (SCC) efficiency of the currently developed g-$C_3N_4$-based photocatalysts for $H_2O_2$ production in pure water is still less than 0.65%[10,13,14,22–24], and the unsatisfactory catalytic activity restricts the industrial production and practical application of photocatalytic $H_2O_2$ synthesis. Therefore, further development of highly active photocatalysts for $H_2O_2$ production in pure water is of great significance and presents challenges.

The photocatalytic activity is considered to be the cumulative result of surface reactions between the active sites on the catalyst and the reactants[25,26]. Several improved strategies (such as introducing surface defects[14,27], doping atoms[18,28,29], designing donor-acceptor units[13,15,30], forming heterojunctions[11,22,31], and developing metal/organic frameworks[32–35], etc.) have been developed to enhance photocatalytic $H_2O_2$ activity. However, due to the structural complexity caused by these strategies, identifying the dynamic structural evolution of active sites in photocatalytic surface reaction and elucidating the corresponding catalytic enhancement mechanism remains a great challenging and, as a result, is rarely reported. In pure water system, understanding how the active sites specifically participate in $O_2$ adsorption and activation during photoactivation at the atomic scale is a fundamental prerequisite for further enhancing the activity of 2e⁻ ORR, which is crucial for the rational development of high-performance 2e⁻ ORR photocatalysts. Excitingly, single-atom photocatalysts (SAPs) with well-defined single-atom active sites and high atomic utilization, serving as idealized catalytic models, provide opportunities for in-depth exploration of the active sites structure evolution and the reaction mechanism[36–38]. Although SAPs have achieved some promising results in 2e⁻ ORR, it still faces the following key problems: (1) Most studies ignore the microscopic structural control of semiconductor substrates in SAPs, resulting in low single-atom loading that cannot provide abundant active sites, making the $H_2O_2$ generation activity in pure water unsatisfactory. (2) More urgently, the dynamic structural evolution of the active sites and corresponding catalytic enhancement mechanism in photocatalytic 2e⁻ ORR under practical reaction conditions remain unclear, severely limiting the further design and development of highly active photocatalysts for $H_2O_2$ synthesis in pure water.

Herein, we report a general method (by tuning the substrate microstructure and optimizing the loading process) for the synthesis of SAPs with high single-atom loading based on g-$C_3N_4$, and successfully synthesize a series of high-loading M-SAPs (M=Fe, Co, Ni, Cu, Zn, Sr, W, Pt) with porous ultrathin structures. Benefiting from the high-concentration single-atom sites exposed by this structure, the developed high-loading Ni single-atom photocatalyst (Ni$_{SAPs}$-PuCN) exhibits high activity and selectivity for $H_2O_2$ generation. Notably, in pure water, the apparent quantum yield (AQY) at 420 nm reaches 10.9% while achieving a high SCC efficiency of 0.82%, which is the most efficient g-$C_3N_4$-based photocatalyst for $H_2O_2$ production reported so far. Pioneeringly, combining in situ synchrotron X-ray absorption spectroscopy, Raman spectroscopy, and theoretical calculation, we directly observed the transformation of initial Ni-$N_3$ sites into high-valent $O_1$-Ni-$N_2$ sites after $O_2$ adsorption during photoactivation. This process promotes the formation of the key intermediate *OOH, which further transforms into HOO-Ni-$N_2$. Crucially, the structure of the $O_1$-Ni-$N_2$ intermediate state ensures the end-on adsorption state of $O_2$ and suitable $O_2$ adsorption energy, leading to a fast transition from ·$O_2^-$ to ·OOH. Overall, this self-optimization of Ni active site evolution (Ni-$N_3$ → $O_1$-Ni-$N_2$ → HOO-Ni-$N_2$) greatly reduces the formation energy barrier

of the intermediate *OOH to accelerate $H_2O_2$ generation ($O_2$ → ·$O_2^-$ → ·OOH → $H_2O_2$), which is the core factor for the high activity and selectivity $H_2O_2$ production of Ni$_{SAPs}$-PuCN. Revealing the catalytic enhancement mechanism through the dynamic structural evolution of active sites provides insights for the development of highly active photocatalysts and a deeper understanding of photocatalysis.

## Results and discussion

### A general synthesis strategy for high-loading M-SAPs

Increasing the loading of single atoms to create more active sites is beneficial to improve the catalytic activity[39,40]. A schematic diagram illustrating the synthesis of high-loading M-SAPs is presented in Fig. 1a, along with the presumed structural changes in the heptazine unit of the corresponding g-$C_3N_4$. Briefly, the synthesis of high-loading M-SAPs is primarily divided into two steps: regulating the microtopography and further optimizing the loading process (involving continuous ultrasonic treatment in wet-chemical precipitation). Firstly, the substrate

microtopography was adjusted by thermal stripping and ultrasonic exfoliation of the original bulk g-$C_3N_4$ (denoted as BCN), thereby preparing porous ultrathin g-$C_3N_4$ nanosheets (denoted as PuCN), which are beneficial for providing more sites for single-atom loading. Next, PuCN was further loaded with metal single atoms by wet-chemical precipitation under continuous ultrasonic conditions. The continuous ultrasonic treatment can not only promote the uniform dispersion of single atoms, but also further exfoliate g-$C_3N_4$ (destroy the van der Waals forces between carbon nitride layers) to provide abundant loading sites, and finally achieve a high-loading M-SAPs with a porous ultrathin structure (denoted as M$_{SAPs}$-PuCN, see "Methods" for details). Importantly, we further confirmed that this synthetic strategy is applicable for the preparation of a series of high-loading M$_{SAPs}$-PuCN (M=Fe, Co, Ni, Cu, Zn, Sr, W, Pt). The aberration-corrected high-angle annular dark-field scanning transmission electron microscopy (HAADF-STEM) images in Fig. 1b–i reveal the uniform dispersion of high-density metal single atoms (Fe, Co, Ni, Cu, Zn, Sr, W, Pt) on g-$C_3N_4$, with no nanoclusters or particles observed. This observation is consistent with the X-ray diffraction (XRD, Supplementary Figs. 1, 2) results of all samples. Energy-dispersive X-ray spectroscopy (EDS, Supplementary Figs. 3–9) mapping further shows that C, N and M elements were uniformly distributed on g-$C_3N_4$. The content of metal loading in M$_{SAPs}$-PuCN was all above 10 wt% by inductively coupled plasma mass spectrometry (ICP-MS, Supplementary Table 1).

### Structural characterization of Ni$_{SAPs}$-PuCN

It was found that Ni$_{SAPs}$-PuCN has the highest photocatalytic $H_2O_2$ generation activity among the prepared various M$_{SAPs}$-PuCN (Supplementary Fig. 10); therefore, the structure of Ni$_{SAPs}$-PuCN was further characterized in detail. The microstructure of the samples was investigated using HAADF-STEM, atomic force microscopy (AFM) and scanning electron microscopy (SEM). In Fig. 2a, BCN exhibited a thick-layered bulk structure, while the PuCN showed a distinct thin layer and porous structure (Fig. 2b). After the ultrasonic-wet chemical loading of Ni single atoms, more abundant pores and thinner undulating folds in Ni$_{SAPs}$-PuCN can be clearly seen (Fig. 2c–d), which are also verified in AFM images (Supplementary Fig. 11). The decrease of the (002) peak intensity of Ni$_{SAPs}$-PuCN in XRD (Supplementary Fig. 2) indicates the weakening of the interlayer stacking, which corresponds to the ultrathin structure. Meanwhile, SEM and TEM (Supplementary Fig. 12), $N_2$ physisorption measurements and pore size distribution (Supplementary Fig. 13 and Supplementary Table 2) together showed that the porous ultrathin structure of Ni$_{SAPs}$-PuCN had a significantly increased specific surface area (139.6 m² g⁻¹, about 8.3 times of BCN) and abundant pore distribution. This adjusted structure would be very favorable for anchoring single atoms, as confirmed in the subsequent HAADF-STEM characterization (Fig. 2e and g). In Fig. 2e, numerous isolated

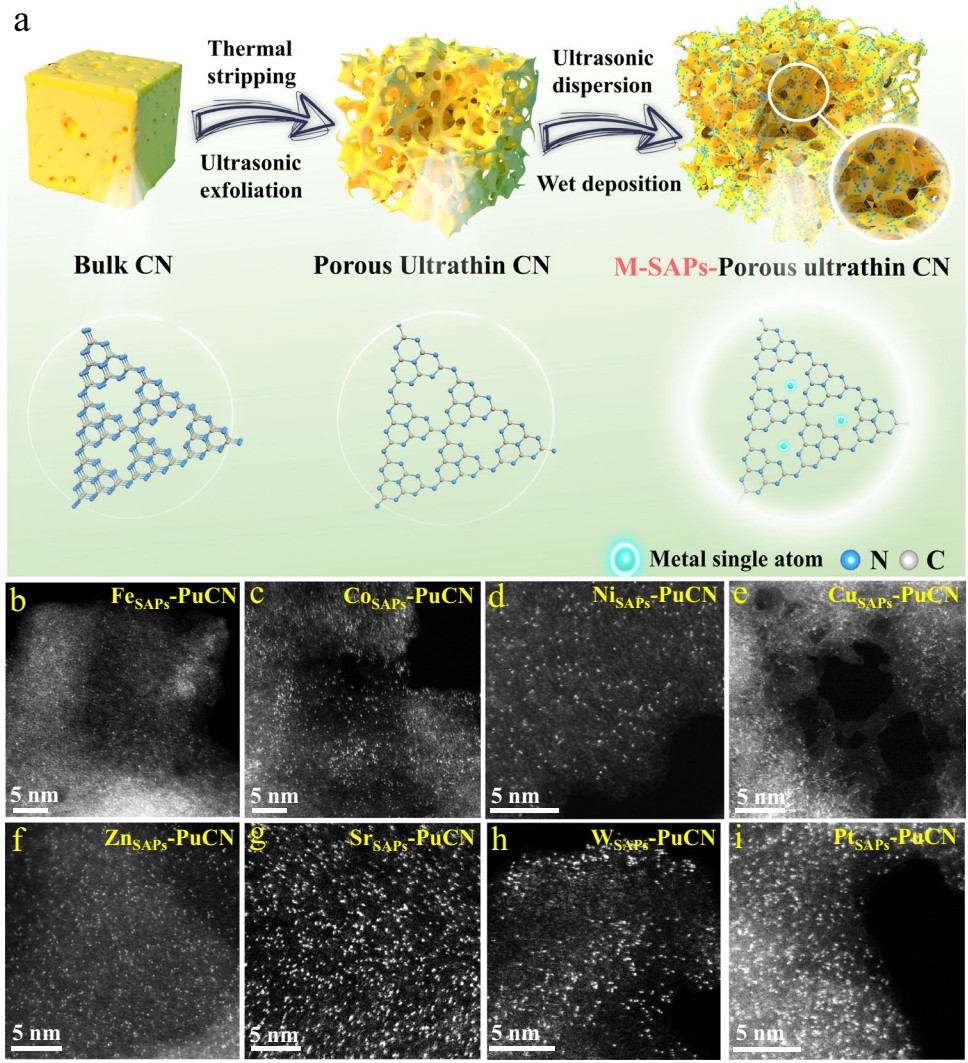

**Fig. 1 | Synthesis and characterization of high-loading single-atom photocatalysts based on g-C₃N₄ (M$_{SAPs}$-PuCN). a** Schematic diagram of the synthesis of high-loading M$_{SAPs}$-PuCN. **b–i** Aberration-corrected HAADF-STEM images of M$_{SAPs}$-PuCN (M = Fe, Co, Ni, Cu, Zn, Sr, W, Pt).

bright spots clearly differ from the g-C₃N₄ substrate contrast (See Supplementary Fig. 14a–c for more details), indicating that the high density of Ni single atoms was successfully dispersed on the Ni$_{SAPs}$-PuCN without any nanoparticles or clusters being found. The EDS mapping (Fig. 2f) showed that Ni, C and N elements were uniformly distributed on the ultrathin porous substrate. As shown in Fig. 2g, high-density Ni single atoms are anchored around the pores of Ni$_{SAPs}$-PuCN, which is favorable for the adsorption and activation of reaction gases in the shuttle pores. The Ni content in Ni$_{SAPs}$-PuCN was as high as 12.5 wt% measured by ICP-MS. In addition, the Fourier transform infrared spectroscopy (FTIR) and X-ray photoelectron spectroscopy (XPS) demonstrated that this synthesis method maintains the structure of g-C₃N₄ (Supplementary Fig. 15). Therefore, based on the above systematic characterizations, high-density Ni$_{SAPs}$-PuCN with porous ultrathin structure was successfully synthesized, and the schematic structure is shown in Fig. 2h.

Furthermore, the coordination structure of Ni in Ni$_{SAPs}$-PuCN was confirmed by X-ray absorption fine structure spectroscopy (XAFS). Figure 3a shows the Ni *K*-edge X-ray absorption near-edge structure (XANES) spectra of Ni$_{SAPs}$-PuCN and Ni foil, NiO, and NiPc as comparisons. The absorption edge position of Ni$_{SAPs}$-PuCN was located between Ni foil and NiO, suggesting that the valence state of Ni in Ni$_{SAPs}$-PuCN was between 0 and +2. The Fourier transform of the

extended X-ray absorption fine structure (FT-EXAFS) spectra of the samples is shown in Fig. 3b. The Ni$_{SAPs}$-PuCN exhibits a main peak around 1.69 Å (Fig. 3b), which is mainly attributed to the scattering interaction between Ni atoms and the first layer (Ni-N). However, no peak of Ni-Ni bond at about 2.17 Å was observed in Ni$_{SAPs}$-PuCN compared to Ni foil. This indicates that Ni species exists in the form of single atoms in Ni$_{SAPs}$-PuCN, validating the results of HAADF-STEM and XRD. Moreover, the wavelet transform (WT) in the Ni *K*-edge EXAFS further analyzes the coordination environment of Ni in Ni$_{SAPs}$-PuCN. Unlike Ni foil and NiO (Fig. 3f and Fig. 3g), the WT contour plot of Ni$_{SAPs}$-PuCN (Fig. 3e) shows that there is only one intensity maximum around 5.23 Å$^{-1}$ (attributed to Ni-N coordination), indicating Ni sites are atomically dispersed on Ni$_{SAPs}$-PuCN. The best-fit analysis results of EXAFS (Fig. 3d and Supplementary Table 3) show that each Ni atom in Ni$_{SAPs}$-PuCN is bonded with three N atoms as Ni-N₃ coordination, and the average Ni-N bond length is about 2.07 Å. The inset of Fig. 3c shows the Ni-N₃ coordination structure model in Ni$_{SAPs}$-PuCN, and the simulated XANES spectra base on this agree well with the experimental results (Fig. 3d), illustrating the rationality of the Ni-N₃ sites. Moreover, further theoretical calculations (Supplementary Fig. 16) also support the fitting results of Ni-N₃. In addition, thermodynamic and kinetic calculation results (Supplementary Fig. 14d–g) clearly demonstrate the rationality and stability of Ni single atoms on g-C₃N₄, consistent with

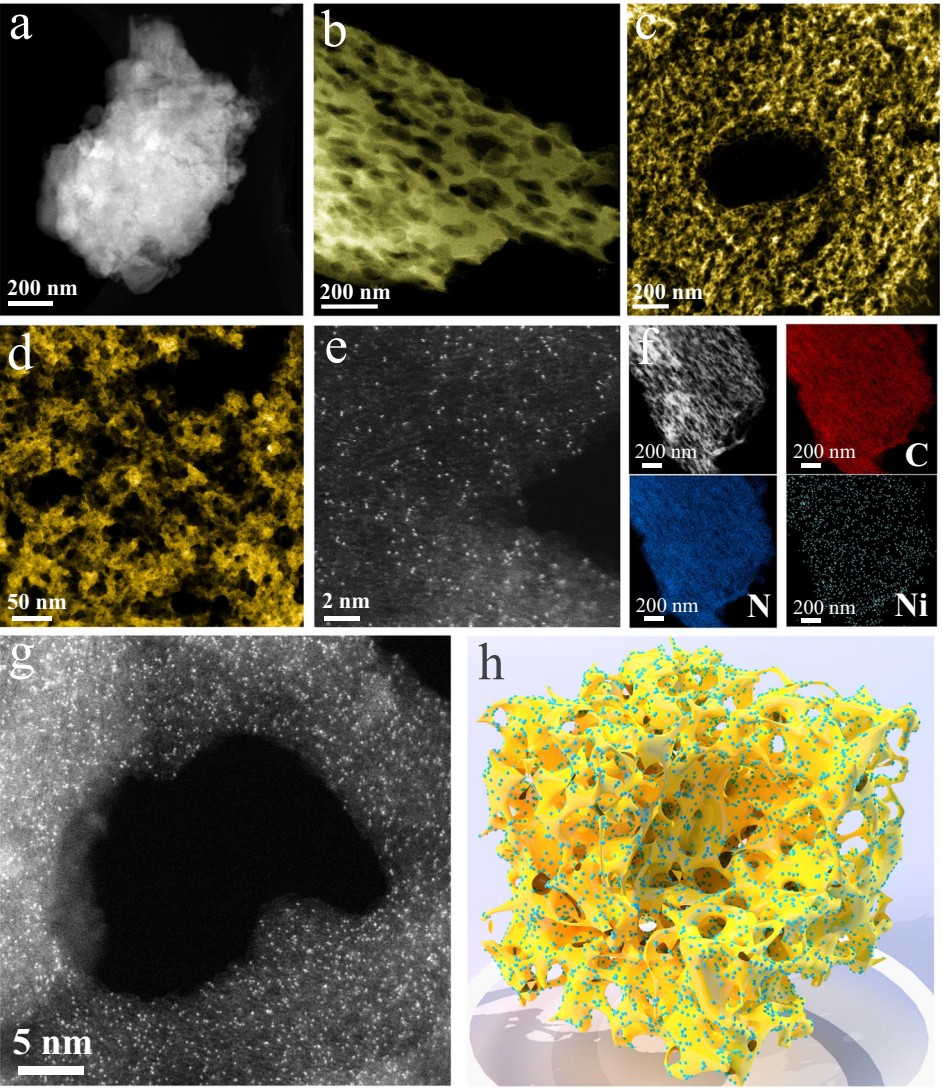

**Fig. 2 | Structural characterization of Ni$_{SAPs}$-PuCN. a–c** HAADF-STEM images of BCN, PuCN, and Ni$_{SAPs}$-PuCN (morphological structure). **d** Local magnification HAADF-STEM image of Ni$_{SAPs}$-PuCN. **e, g,** Aberration-corrected HAADF-STEM images of Ni$_{SAPs}$-PuCN at different positions. **f** EDS mapping images of Ni$_{SAPs}$-PuCN. **h** Schematic structure of Ni$_{SAPs}$-PuCN.

experimental characterization results, in which Ni atoms exist as isolated single atoms rather than in the form of clusters.

## Catalytic activity and selectivity of Ni$_{SAPs}$-PuCN for photocatalytic H$_2$O$_2$ production

The photocatalytic H$_2$O$_2$ generation performance of the samples was evaluated in O$_2$-saturated pure water (without any sacrificial agent or buffer salt solution; pH = 7) under visible light irradiation ($\lambda \geq 420$ nm). As shown in Fig. 4a, the H$_2$O$_2$ generation rate of PuCN (41.1 μmol L$^{-1}$ h$^{-1}$) was slightly increased relative to BCN (16.5 μmol L$^{-1}$ h$^{-1}$), which could be the porous ultrathin structure expanded the contact range of O$_2$. Notably, the Ni$_{SAPs}$-PuCN exhibited significantly enhanced H$_2$O$_2$ generation rate up to 342.2 μmol L$^{-1}$ h$^{-1}$ (Fig. 4a), which was 20.7 and 8.3 times of BCN and PuCN, respectively (Fig. 4b). This indicates that the Ni single atoms are the core of the enhanced H$_2$O$_2$ production activity. Meanwhile, the effect of Ni single atoms loading content and Ni nanoparticles on the H$_2$O$_2$ activity was further sorted out (Supplementary Figs. 17, 18). Furthermore, the H$_2$O$_2$ generation activity of Ni$_{SAPs}$-PuCN can be improved to 640.1 μmol L$^{-1}$ h$^{-1}$ under AM 1.5 G irradiation (Fig. 4b). The variable comparison and radical trapping experiments.

(Supplementary Fig. 19) confirm that H$_2$O$_2$ is indeed generated by the 2e$^-$ ORR pathway. The oxidation reaction of photogenerated holes was confirmed by photocatalytic O$_2$ production test (Supplementary Fig. 20), which is consistent with previous reports[10]. The final yield of photocatalytic H$_2$O$_2$ production depends on the formation and decomposition rates of H$_2$O$_2$, and the experiment (Supplementary Fig. 21) suggested that the decomposition rate of H$_2$O$_2$ on the samples is relatively slow. A systematic performance comparison of Ni$_{SAPs}$-PuCN with recently reported g-C$_3$N$_4$-based materials and other types of materials for photocatalytic H$_2$O$_2$ production in pure water is presented in Supplementary Table 4. The comparison of corresponding normalized H$_2$O$_2$ yields (μmol g$^{-1}$ h$^{-1}$) in Supplementary Fig. 22 demonstrates the efficient H$_2$O$_2$ generation activity of Ni$_{SAPs}$-PuCN.

To further evaluate the light utilization efficiency of Ni$_{SAPs}$-PuCN in pure water, the AQY was measured under monochromatic light irradiation (Fig. 4c and Supplementary Table 5). The AQY of Ni$_{SAPs}$-PuCN at 420 nm reaches 10.9%, surpassing most of reported g-C$_3$N$_4$-based photocatalysts and currently developed materials (Supplementary Fig. 23 for AQY comparison and Supplementary Table 4). Moreover, in Fig. 4f, the SCC efficiency of Ni$_{SAPs}$-PuCN can reach 1.17% in the first hour and finally stabilized at 0.82% in 3 h, which is the

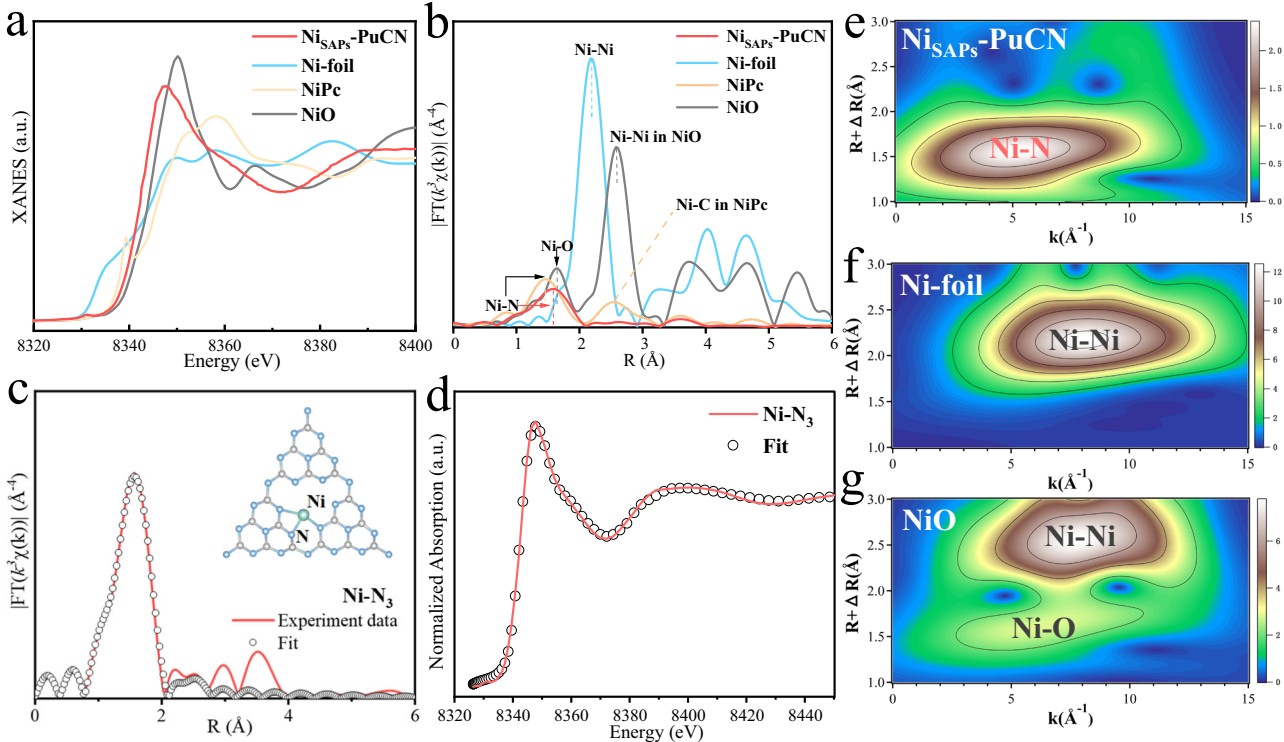

**Fig. 3 | Ni single atoms coordination structure characterization in Ni$_{SAPs}$-PuCN.**
**a** Ni $K$-edge XANES spectra of Ni$_{SAPs}$-PuCN, Ni-foil, NiO, and NiPc. **b** Fourier transformation of the EXAFS spectra. **c** First-shell (Ni−N) fitting of Fourier transformations of EXAFS spectra for Ni$_{SAPs}$-PuCN. **d** Simulated XANES spectra based on Ni-N$_3$ model for DFT calculations. **e**–**g** Wavelet transform EXAFS spectra of Ni$_{SAPs}$-PuCN, Ni-foil, and NiO, respectively.

highest value reported so far for g-C$_3$N$_4$-based photocatalysts in pure water (Fig. 4g for SCC efficiency comparison and Supplementary Table 4). Simultaneously, in Fig. 4g, when compared with other types of photocatalysts currently developed, the SCC efficiency of Ni$_{SAPs}$-PuCN still demonstrates advantages and approaches the highest SCC efficiency values (1.0%–1.2%) reported for powder photocatalyst[15,30]. Overall, according to the standard H$_2$O$_2$ yield, AQY and SCC efficiency as the evaluation indicators of photocatalytic activity, Ni$_{SAPs}$-PuCN has excellent H$_2$O$_2$ generation activity in pure water. Meanwhile, Ni$_{SAPs}$-PuCN exhibited excellent performance cyclability (Supplementary Fig. 24) and structural stability (Supplementary Fig. 25a–f). The thermodynamic stability of the Ni$_{SAPs}$-PuCN model was also confirmed by molecular dynamics (MD) calculations (Supplementary Fig. 25g).

The high ORR selectivity is the guarantee of high activity[2,4]. The effect of Ni single atoms on O$_2$ selective electron transfer was investigated by electrochemical measurements on rotating ring-disk electrode (RRDE), where the disk current was derived from the reduction reaction of O$_2$, while the ring current was derived from the oxidation reaction of the produced H$_2$O$_2$. The selectivity of H$_2$O$_2$ generation on BCN and Ni$_{SAPs}$-PuCN was monitored in O$_2$-saturated 0.1 M KOH electrolyte. In Fig. 4d (bottom), the reduction disk currents of PCN and Ni$_{SAPs}$-PuCN gradually increased as the potential decreased from 0.8 V (vs. RHE). In Fig. 4d (top), the ring current of Ni$_{SAPs}$-PuCN was significantly higher than that of BCN, indicating that Ni$_{SAPs}$-PuCN produced more H$_2$O$_2$. As shown in Fig. 4e, the average number of transferred electrons (n) and H$_2$O$_2$ selectivity were calculated in the potential range of 0–0.6 V (vs. RHE). Under the same conditions, the number of transferred electrons of Ni$_{SAPs}$-PuCN is closer to 2 and the selectivity of H$_2$O$_2$ is higher than that of BCN. The number of electrons transferred on BCN was 2.77, and the H$_2$O$_2$ selectivity was only 61.4% at 0.5 V (vs. RHE). But at the same potential, the number of electrons transferred on Ni$_{SAPs}$-PuCN is 2.14, achieving 92.4% H$_2$O$_2$ selectivity. Combined with the RRDE and photocatalytic H$_2$O$_2$ performance tests, these fully demonstrate that Ni single-atom sites greatly improve the 2e⁻ selectivity of O$_2$ and efficiently promote H$_2$O$_2$ generation.

## Electronic Structure and carrier separation properties of Ni$_{SAPs}$-PuCN

The electronic structure of photocatalysts largely determines the carrier separation characteristics and further affects the surface reaction efficiency[19,26,41,42]. In the UV-Vis diffuse reflectance spectroscopy (DRS) of the samples (Supplementary Fig. 26a), the introduction of Ni single atoms expands the absorption of visible light and adjusts the band structure. The introduction of Ni single atoms effectively promotes carrier separation and transport, as confirmed by a series of spectroscopic measurements (Supplementary Fig. 27). Next, femtosecond transient absorption spectroscopy (fs-TAS) was used to further investigate the kinetic behaviors of the photogenerated carriers. The results indicate that Ni$_{SAPs}$-PuCN has a shorter lifetime than BCN (See Supplementary Fig. 28 for more details), which could be attributed to the deep trapping sites induced by Ni single atoms and has been demonstrated to facilitate the 2e⁻ ORR process[10]. Bader charge analysis (Supplementary Fig. 29) indicates that the chemical valence of Ni single-atom in Ni$_{SAPs}$-PuCN is positive, which is in line with the results of Ni 2$p$ XPS (Supplementary Fig. 15d) and XANES (Fig. 3a). The total density of states (TDOS) of BCN exhibits a typical semiconducting nature (Supplementary Fig. 30a), where the valence band (VB) are mainly composed of N 2$p$ orbitals, while the conduction band (CB) is mainly composed of C 2$p$ and N 2$p$ orbitals[43,44]. As for Ni-$_{SAPs}$-PuCN, the introduction of Ni atoms creates impurity levels and narrows the band gap (Supplementary Fig. 30b), which agrees well with the experimental results (Supplementary Fig. 26b). Combined with the projected density of states (PDOS) of Ni (Supplementary Fig. 31), the Ni 3$d$ orbitals contribute to both CB and VB, implying that Ni single atoms greatly optimize the electronic structure. Further, in the charge density difference of Ni$_{SAPS}$-PuCN (Supplementary Fig. 32), there are remarkably

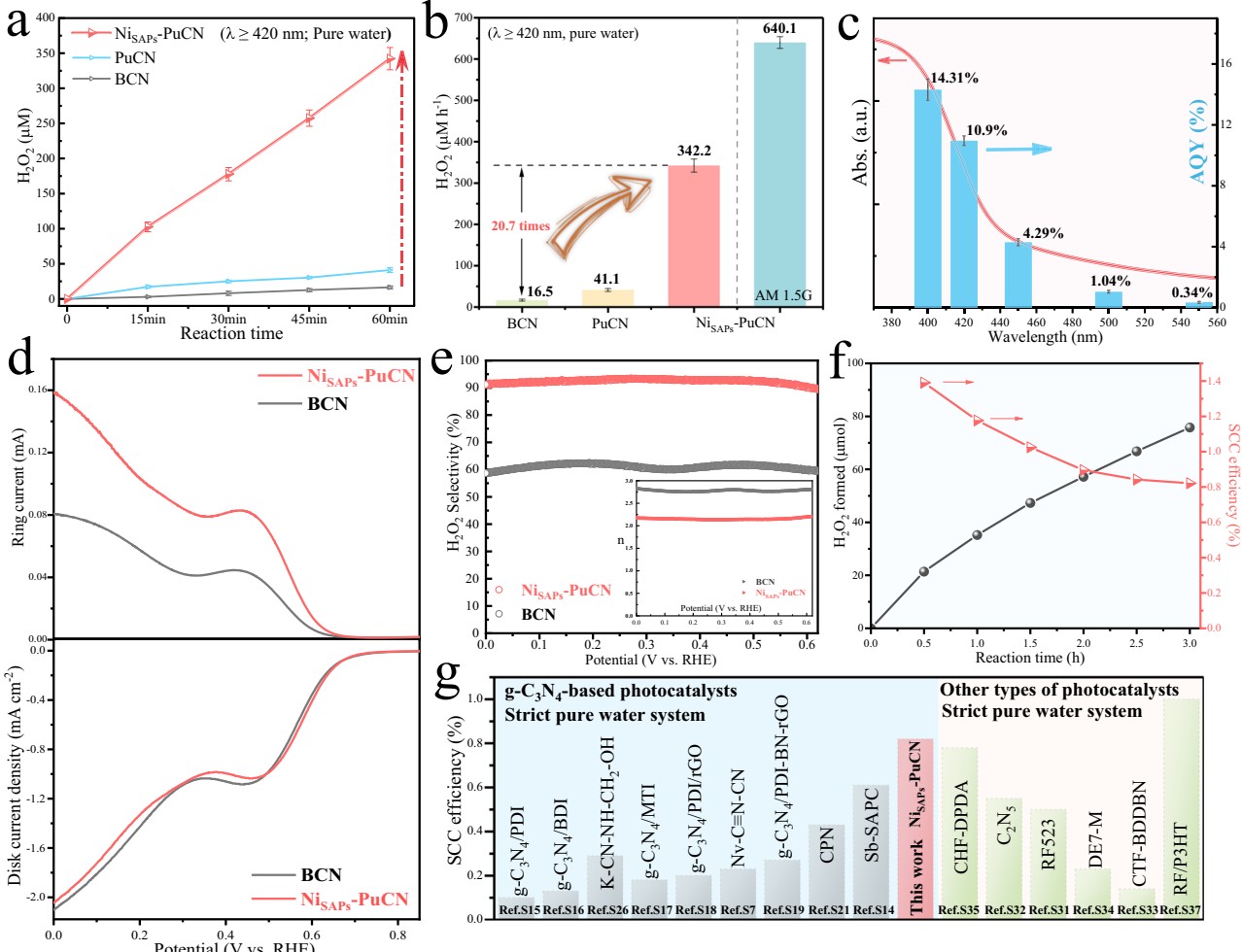

**Fig. 4 | Photocatalytic H₂O₂ production activity and selectivity of Ni_SAPs-PuCN.**
**a** The time course of H₂O₂ production measured in pure water under visible light irradiation ($\lambda \geq 420$ nm, 60 mW cm⁻²; 30 mg catalyst in 30 ml pure water, 1 g L⁻¹ catalyst; 25 °C). Error bars are the standard deviations of three replicate measurements. **b** Performance comparison of all samples in pure water under visible light conditions and H₂O₂ generation performance of Ni_SAPs-PuCN under AM 1.5 G illumination (60 mW cm⁻²). **c** The wavelength-dependent AQY for photocatalytic H₂O₂ production in pure water by Ni_SAPs-PuCN (50 mg catalyst in 50 ml pure water, 1 g L⁻¹ catalyst, 25 °C). **d** RRDE polarization curves over BCN and Ni_SAPs-PuCN in O₂-saturated 0.1 M KOH at 1600 rpm with ring current (upper part) and disk current (bottom part). **e** H₂O₂ selectivity as a function of the applied potential. The inset shows the calculated average number of transferred electrons (n). **f** The amount of H₂O₂ generated by Ni_SAPs-PuCN under AM 1.5 G simulated sunlight irradiation (100 mW cm⁻²) and the corresponding SCC efficiency (500 mg in 100 ml pure water, 25 °C). **g** Summarized SCC efficiencies of recently reported photocatalysts (g-C₃N₄-based and other types of photocatalysts) for H₂O₂ production in pure water.

charge redistribution between Ni single-atom and N atoms, which would help to facilitate the separation and transport of photogenerated carriers. The optimized carrier separation and transport properties of Ni_SAPs-PuCN will facilitate subsequent efficient surface reactions.

**In situ XAFS analysis of Ni-N₃ site evolution during photoactivation**
To gain deep insight into the structural evolution of Ni sites and catalytic enhancement mechanism, in situ XAFS measurements were performed to monitor the details of O₂ adsorption and activation on Ni single atoms at the atomic scale. Figure 5a–b shows the normalized Ni K-edge XANES spectra and the corresponding FT-EXAFS spectra. As shown in Fig. 5a, in the Ar-saturated aqueous solution, the Ni single atoms in Ni_SAPs-PuCN were still Ni-N₃ coordinated (Supplementary Fig. 33 and Table 6), indicating that the aqueous solution does not affect the Ni coordination structure. However, the white line intensity of the Ni K-edge XANES spectra was clearly enhanced in O₂-saturated aqueous solution compared to the Ar-saturated aqueous solution (enlarged view of Fig. 5a). This indicates an increase in Ni oxidation

state, possibly due to the delocalization of unpaired electrons in Ni 3d orbitals and the spontaneous charge transfer from Ni to the O 2p orbitals of O₂, which promotes the formation of superoxide radicals (·O₂⁻). Next, EPR experiments and theoretical calculations can provide evidence for our analysis. On the one hand, the ·O₂⁻, as a key free radical for photocatalytic 2e⁻ ORR, is formed from the electron obtained by the activated O₂ (O₂ + e⁻ → ·O₂⁻)[10,14]. The EPR trapping experiments further confirmed the presence of ·O₂⁻. As shown in Fig. 5f, compared with the dark condition, Ni_SAPs-PuCN under the light condition had a stronger ·O₂⁻ signal, corresponding to Ni losing electrons (resulting in an increased oxidation state) and transferring electrons to O₂ to generate ·O₂⁻. On the other hand, based on theoretical calculations, the optimized structure and charge difference density of Ni_SAPs-PuCN after adsorption of O₂ are shown in Fig. 5d, which can intuitively reflect the charge transfer from Ni sites to the end-on adsorbed O₂ (O₁-Ni-N₂). Detailed theoretical calculations illustrated the plausibility of O₁-Ni-N₂ after O₂ adsorption (Supplementary Fig. 34). Furthermore, this charge transport mechanism was confirmed again based on Bader charge analysis, where the adsorbed O₂ molecules gain electrons (0.48|e|) from Ni_SAPs-PuCN (Fig. 5d). Based on the above results, to further

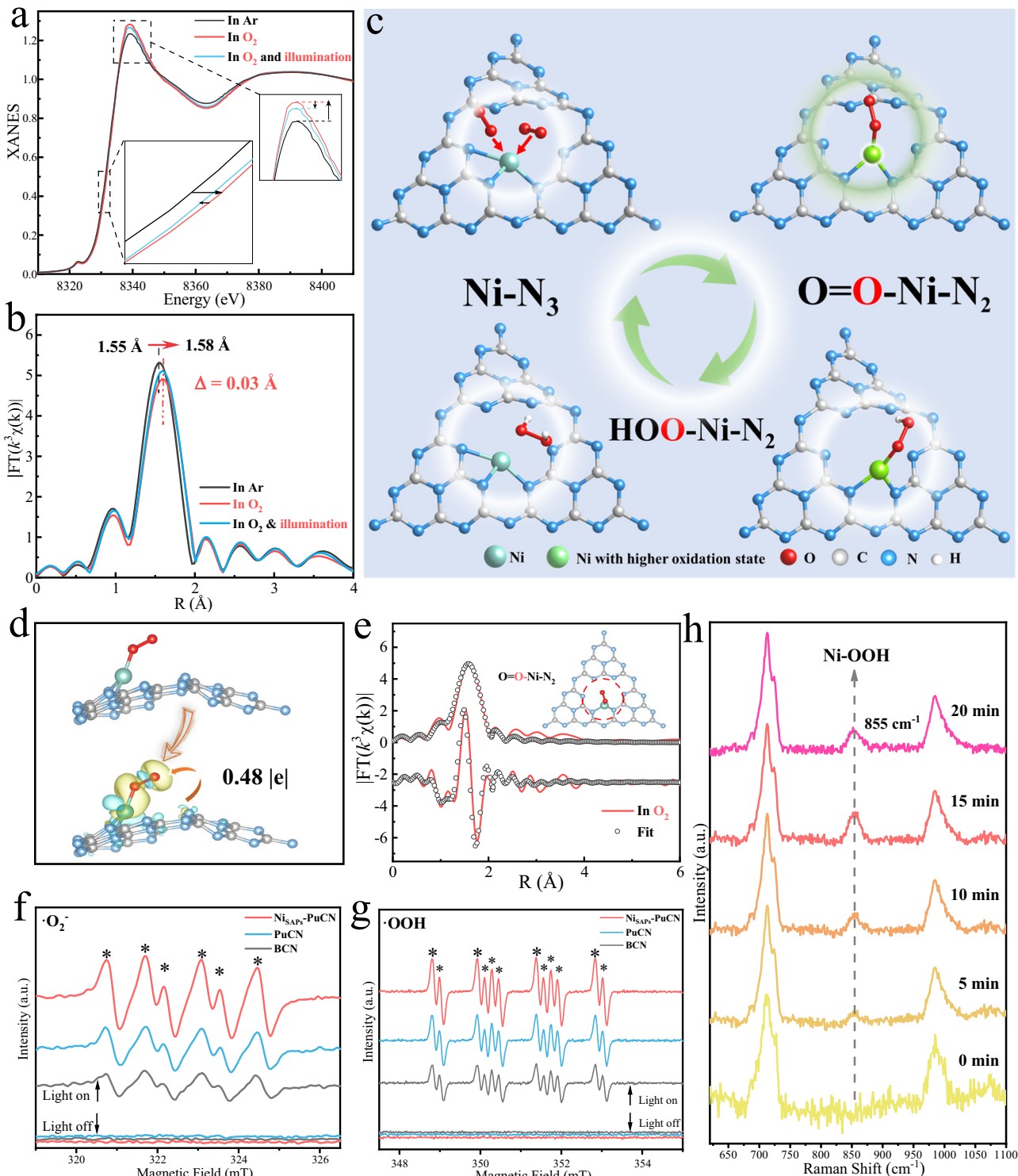

**Fig. 5 | In situ structural evolution of Ni sites in photocatalytic 2e⁻ ORR. a** Ni *K*-edge XANES spectra of Ni$_{SAPs}$-PuCN during photocatalytic 2e⁻ ORR in Ar or O$_2$ saturated aqueous solution at room temperature. The inset is the enlarged Ni *K*-edge XANES spectra. **b** FT-EXAFS spectra of Ni$_{SAPs}$-PuCN under in situ operation. **c** Schematic diagram of Ni sites structure evolution of Ni$_{SAPs}$-PuCN in the photocatalytic 2e⁻ ORR. **d** Optimized structures and charge difference density of adsorbed O$_2$ molecule on Ni$_{SAPs}$-PuCN (The isosurface value is 0.0016 eV Å⁻³, electron accumulation and consumption are indicated in yellow and blue, respectively). **e** First-shell fitting of FT-EXAFS spectra for Ni$_{SAPs}$-PuCN in O$_2$ saturated aqueous solution. **f, g** EPR signals of ·O$_2$⁻ and ·OOH of the samples in the presence of DMPO. **h** Raman spectra of Ni$_{SAPs}$-PuCN recorded during the photoreaction in O$_2$-saturated aqueous solution.

elucidate the local coordination structure evolution of the Ni active sites in O$_2$, the Ni *K*-edge EXAFS fitting results (Fig. 5e and Supplementary Table 6) in O$_2$-saturated solution revealed additional Ni-O coordination (bond length about 2.10 Å), fitted as O$_1$-Ni-N$_2$

coordination (schematic inset of Fig. 5e). The above results comprehensively demonstrate that the active sites of Ni$_{SAPs}$-PuCN evolved from the initial Ni-N$_3$ to O$_1$-Ni-N$_2$ at the O$_2$ adsorption and activation stages.

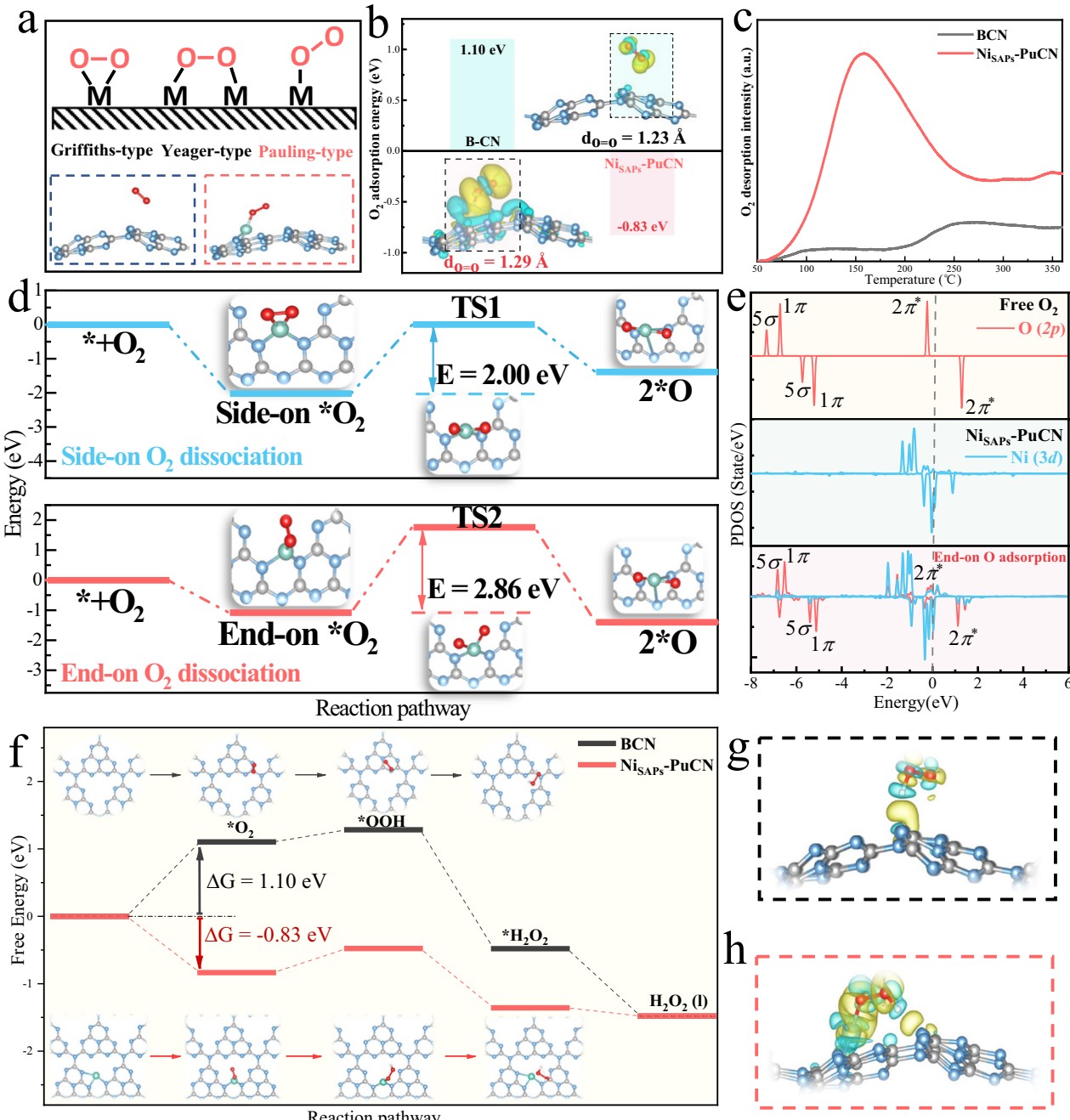

**Fig. 6 | Catalytic enhancement mechanism of Ni sites evolution in Ni$_{SAPs}$-PuCN.**
**a** Schematic diagram of O$_2$ adsorption structure on metal sites (upper part), and optimized structures of adsorbed O$_2$ molecule on BCN and Ni$_{SAPs}$-PuCN (bottom part). **b** Comparison of O$_2$ adsorption energy and charge difference density on BCN and Ni$_{SAPs}$-PuCN. **c** Temperature programmed O$_2$ desorption (TPD-O$_2$) profiles of BCN and Ni$_{SAPs}$-PuCN. **d** O$_2$ dissociation energies and profile corresponding to Ni$_{SAPs}$-PuCN side-on (upper part) and end-on O$_2$ (bottom part) adsorption

configurations. **e** The PDOS plots for free O$_2$, Ni$_{SAPs}$-PuCN and Ni$_{SAPs}$-PuCN after end-on O$_2$ adsorption. **f** Free energy profiles for photocatalytic H$_2$O$_2$ evolution reactions over BCN and Ni$_{SAPs}$-PuCN. Optimized models and charge difference density after BCN (**g**) and Ni$_{SAPs}$-PuCN (**h**) generate *OOH (The isosurface value is 0.0016 eV Å$^{-3}$. Electron accumulation and consumption are indicated in yellow and blue, respectively).

Next, as shown in Fig. 5a, when light sources are introduced to initiate the photocatalytic 2e$^-$ ORR, the Ni $K$-edge shifts to the lower energy direction (between the O$_2$-saturated aqueous solution and the Ar-saturated aqueous solution), corresponding to a decrease in the oxidation state. This suggests that the 2e$^-$ ORR reaction is further driven (Ni(*) + O$_2$ + 2e$^-$ + 2H$^+$ → Ni (*) + H$_2$O$_2$: O$_2$ → ·O$_2^-$ → ·OOH → H$_2$O$_2$). Nonetheless, it is difficult to accurately capture the key intermediate *OOH on the Ni site using in situ XAFS, so the Raman spectroscopy (Fig. 5h) was used to further identify the reaction details. In Fig. 5h,

Ni$_{SAPs}$-PuCN gradually exhibits a new absorption band at 855 cm$^{-1}$ in O$_2$ aqueous solution with illumination time, which can be attributed to the O=O stretching mode in Ni-OOH[10,45,46]. Moreover, theoretical calculation (Supplementary Fig. 35) further reveal that the *OOH on the Ni sites is in an end-on adsorption configuration (HOO-Ni-N$_2$, Fig. 5c and Fig. 6h). Simultaneously, the ·OOH radicals were also detected through EPR test[30,47]. In Fig. 5g, compared with BCN and PuCN, Ni$_{SAPs}$-PuCN had a stronger ·OOH signal under the same conditions, implying that Ni single atoms can effectively promote the ·OOH generation.

Throughout the in situ XAFS, the FT-EXAFS spectra (Fig. 5b) show that the main peak shifts (-0.03 Å) to longer lengths, implying the expansion of the Ni-N bonds during the reaction. Thus, combined with the above comprehensive analysis, we elucidated the structural evolution mechanism of Ni active sites on Ni$_{SAPs}$-PuCN in photocatalytic 2e$^-$ ORR, and the corresponding schematic diagram is shown in Fig. 5c. In photocatalytic 2e$^-$ ORR, after O$_2$ adsorption and activation, the Ni-N$_3$ sites are transformed into O$_1$-Ni-N$_2$, which further promotes the formation of the key *OOH intermediate with end-on adsorption configuration (HOO-Ni-N$_2$), thereby accelerating the generation of H$_2$O$_2$.

**In-depth exploration of Ni site structure evolution and catalytic enhancement mechanism**

To further understand the relationship between structural evolution of Ni active sites and high H$_2$O$_2$ activity, we systematically investigated the surface reaction mechanism of 2e$^-$ ORR by combining theoretical calculations and experiments. The first step of the 2e$^-$ ORR surface reaction is O$_2$ adsorption and activation. The O$_2$ adsorption configuration and the O$_2$ adsorption energy on the catalyst are particularly critical for the subsequent reactions[48,49]. The adsorption configuration of O$_2$ on the catalyst is categorized into three basic types (Fig. 6a, upper): Yeager type (side-on), Griffith type (side-on), and Pauling type (end-on)[48,50]. The end-on O$_2$ adsorption configuration, which tends to maintain the O=O bond, can inhibit the 4e$^-$ ORR (O$_2$ + 4e$^-$ + 4H$^+$ → 2H$_2$O) and promote the highly selective 2e$^-$ ORR[7,48,49,51]. Based on this, we first performed first-principles calculations to investigate the O$_2$ adsorption and activation properties in this system. As shown in the calculation results in Fig. 6a (bottom), in comparison to BCN, O$_2$ was adsorbed by Ni single atoms on Ni$_{SAPs}$-PuCN in an end-on configuration (O$_1$-Ni-N$_2$), which is consistent with the fitted data from in situ XAFS (Fig. 5e). Simultaneously, the O$_2$ adsorption Gibbs free energy on Ni$_{SAPs}$-PuCN (-0.83 eV, Fig. 6b) was much lower than that of BCN (1.10 eV, Fig. 6b), indicating that Ni single atoms can efficiently adsorb O$_2$. The charge difference density after O$_2$ adsorption on the two samples (inset of Fig. 6b) was further calculated and analyzed. Compared to BCN (the bond length of adsorbed O$_2$ on BCN is 1.23 Å), the adsorbed O$_2$ on Ni$_{SAPs}$-PuCN has a longer O=O bond length (1.29 Å) and a larger charge transfer, indicating that the O$_1$-Ni-N$_2$ intermediate structure promotes the O$_2$ adsorption and activation. Furthermore, temperature-programmed desorption of O$_2$ (TPD-O$_2$, Fig. 6c) suggested that Ni$_{SAPs}$-PuCN has remarkable stronger O$_2$ adsorption capacity than BCN, which fully confirms our calculation results.

In addition, maintaining the O=O bond of O$_2$ and avoiding the dissociation of O$_2$ are important prerequisites for H$_2$O$_2$ generation in 2e$^-$ ORR[4,52]. Different O$_2$ adsorption configurations affect the extent of O$_2$ activation and the subsequent reactions, and these effects were investigated through theoretical calculations. As shown in Fig. 6d, the O$_2$ dissociation barriers for side-on (O$_2$-Ni-N$_2$) and end-on O$_2$ adsorption configurations (O$_1$-Ni-N$_2$) on Ni$_{SAPs}$-PuCN are 2.00 eV and 2.86 eV, respectively. This indicates that the side-on adsorbed O$_2$-Ni-N$_2$ is more inclined to dissociate O$_2$, while the end-on adsorbed O$_1$-Ni-N$_2$ intermediate structure inhibits the dissociation of O$_2$ and favors the H$_2$O$_2$ generation, further emphasizing the importance of O$_1$-Ni-N$_2$ in the structural evolution. The PDOS was further used to elucidate the charge transfer between the end-on adsorbed O$_2$ molecule and Ni$_{SAPs}$-PuCN. As shown in Fig. 6e, when free O$_2$ with a spin triplet ground state is adsorbed onto Ni$_{SAPs}$-PuCN, the 2$p$ orbitals of O exhibit noticeable hybridization with the Ni 3$d$ orbitals of Ni$_{SAPs}$-PuCN. The O$_2$ molecule donates the $p$-electron to the empty 3$d$ orbital of the Ni atom, and then the 3$d$ electron from the Ni atom is donated back to the 2$\pi^*$ orbital of O$_2$. As a result, the original empty 2$\pi^*$ orbitals in the spin-down channel of free O$_2$ are partially occupied, leading to the elongation of the O=O bond length. Based on the above analysis, the improved O$_2$ adsorption and activation properties of Ni active sites provide strong support for

the formation of O$_1$-Ni-N$_2$ sites in in situ XAFS. Moreover, the critical intermediate state with the O$_1$-Ni-N$_2$ structure ensures suitable O$_2$ adsorption energy and adsorption configuration, inhibits O$_2$ dissociation, and improves the selectivity, thus guaranteeing the subsequent efficient H$_2$O$_2$ generation.

Based on the above results, the free energy changes of the photocatalytic 2e$^-$ ORR on the BCN and Ni$_{SAPs}$-PuCN were further calculated. In Fig. 6f, the introduction of Ni active sites greatly facilitates the O$_2$ adsorption compared to BCN (reduced from 1.10 eV to -0.83 eV, and the theoretical calculation of BCN for O$_2$ adsorption is in Supplementary Fig. 36). The formation and hydrogenation of the core intermediate *OOH are also the key to 2e$^-$ ORR[10,14]. The enhanced O$_2$ adsorption capacity of Ni$_{SAPs}$-PuCN further facilitates *OOH formation, thus evidently promoting the conversion of *OOH to H$_2$O$_2$. Moreover, Fig. 6g–h shows the charge density difference between the two structures and the resulting *OOH. Contrast to BCN, the charge redistribution between *OOH and Ni$_{SAPs}$-PuCN is more remarkable, reflecting that Ni sites effectively promote the generation of key intermediate *OOH, which is also in line with the EPR test (Fig. 5g). Combined with the ·O$_2^-$ and ·OOH radical tests of EPR (Fig. 5f–g), the highly active Ni$_{SAPs}$-PuCN has the strongest ·O$_2^-$ and ·OOH signals in contrast, indicating that the evolution of the Ni site structure is accompanied by a rapid transformation mechanism from ·O$_2^-$ to ·OOH, which strongly supports the above calculations. More importantly, theoretical simulations also confirm that the structural evolution of Ni sites (structure diagram in Fig. 6f) during this process is consistent with in situ experiments: from Ni-N$_3$ to O$_1$-Ni-N$_2$ to HOO-Ni-N$_2$, corresponding to the reaction mechanism from O$_2$ to ·O$_2^-$ to ·OOH to H$_2$O$_2$. This dynamic structural evolution mechanism of Ni single atoms represents the self-optimization of active sites in 2e$^-$ ORR surface reaction, which is the core factor for Ni$_{SAPs}$-PuCN with high activity and high selectivity for H$_2$O$_2$.

In summary, we present a general synthesis strategy for high-loading M$_{SAPs}$-PuCN (M = Fe, Co, Ni, Cu, Zn, Sr, W, Pt) with porous ultrathin structure. This approach can be applied to various catalytic reactions and energy conversion fields. The well-designed high-loading Ni$_{SAPs}$-PuCN exhibits excellent photocatalytic H$_2$O$_2$ performance, with Ni single atoms optimizing the electronic structure and providing numerous highly active reaction sites. Importantly, through in situ XAFS and theoretical calculations, we reveal the structural evolution of Ni single-atom active sites (Ni-N$_3$ → O$_1$-Ni-N$_2$ → HOO-Ni-N$_2$) in surface reactions, closely related to the high catalytic activity. The O$_1$-Ni-N$_2$ intermediate state structure ensures the proper O$_2$ adsorption configuration and energy, not only suppressing O$_2$ dissociation and enhancing the selectivity, but also promoting intermediate conversion to H$_2$O$_2$. This dynamic self-tuning of the coordination structural evolution significantly lowers the formation energy barrier of *OOH, which is a pivotal factor contributing to the high activity and selectivity of Ni$_{SAPs}$-PuCN. Elucidating the mechanism of structural evolution closely associated with high catalytic activity paves the way for the design of highly efficient photocatalysts and a deeper comprehension of photocatalysis.

## Methods
### Synthesis of BCN
10 g of urea was placed in a covered alumina crucible and heated to 550 °C at a rate of 5 °C min$^{-1}$ in a muffle furnace for 2 h. After the crucible was cooled to room temperature, the product was collected and ground into powder, washed several times with deionized water, and dried under vacuum at 80 °C for 12 h (the obtained BCN was about 300 mg).

### Synthesis of PuCN
The PuCN was obtained by thermal stripping and ultrasonic stripping of BCN. Similar to the synthesis of BCN, the holding time at 550 °C in the

muffle furnace was extended to 4 h. After cooling, 50 mg of the powder was ultrasonically stripped in 100 ml of pure water for 8 h and collected by centrifugation. The product was drop-coated evenly in glassware and dried overnight in vacuum (the obtained PuCN was about 45 mg).

### Synthesis of Ni$_{SAPs}$-PuCN and various M$_{SAPs}$-PuCN

Put 50 mg of PuCN in 40 ml of pure water and add 10 ml of ethanol, first sonicate for 2 h to evenly disperse the sample. Then, under continuous ultrasonic and stirring, a certain amount of NiCl$_2$·6H$_2$O solution (45 mg of NiCl$_2$·6H$_2$O in 30 ml of solvent, 1.5 mg ml$^{-1}$; water to ethanol in the solvent is 1:1) was slowly added dropwise, and ultrasonically treated for 3 h to ensure full contact between metal ions and PuCN (continuous sonication is beneficial to promote uniform dispersion and high loading of single atoms, see Supplementary Fig. 37 for more details). The volume of Ni solution added (5 ml, 10 ml or 15 ml, etc.) is controlled to adjust the metal loading. Then the mixed solution was heated in an oil bath at 60 °C for 4 h with vigorous stirring. The product was collected by centrifugation and dried overnight at 80 °C in a vacuum oven. The powder was fully ground and heated to 350 °C in an Ar atmosphere at 2 °C min$^{-1}$ in a tube furnace and kept for 2 h, and finally cooled to room temperature. The collected powders were washed multiple times with 2% (v/v) HCl to remove nanoparticles or clusters, then washed at least 3 times with pure water and dried overnight in a vacuum oven at 80 °C to obtain Ni$_{SAPs}$-PuCN (about 40 mg). The maximum Ni single-atom loading in Ni$_{SAPs}$-PuCN is 12.5 wt %, and more details are shown in Supplementary Fig. 17 and Supplementary Table 1. The synthesis method of M$_{SAPs}$-PuCN is the same as that of Ni$_{SAPs}$-PuCN. Different metal salt solutions, such as FeCl$_3$·6H$_2$O, CoCl$_2$·6H$_2$O, CuCl$_2$, ZnCl$_2$, SrCl$_2$, Na$_2$WO$_4$·2H$_2$O and H$_2$PtCl$_6$, were used to prepare M$_{SAPs}$-PuCN (M = Fe, Co, Cu, Zn, Sr, W, Pt).

### Photocatalytic H$_2$O$_2$ production reaction

30 mg of photocatalyst was placed in 30 ml of pure water (pH = 7), and ultrasonically treated for 30 min until the catalyst powder was completely dispersed. The solution was continuously filled with O$_2$ for 1 h in the dark to saturate the O$_2$. The reaction solution was irradiated with a 420 nm cut-off film (λ ≥ 420 nm) under a 300 W Xe lamp (PLS-SXE 300D/DUV, PerfectLight) and started the photoreaction test under magnetic stirring. The flow rate of the O$_2$ was 50 ml min$^{-1}$ and the reaction temperature was controlled at 25 °C by circulating water. The average light intensity was 60 mW cm$^{-2}$. Every 15 min, 2 ml of the reaction solution was taken out, and the photocatalyst was removed by centrifugal filtration. The H$_2$O$_2$ content generated by the reaction was detected by iodometric method[14,28]. Briefly, 0.5 ml of 0.4 mol L$^{-1}$ potassium iodide (KI) solution and 0.5 ml of 0.1 mol L$^{-1}$ potassium hydrogen phthalate (C$_8$H$_5$KO$_4$) solution was added to 1 ml obtained solution and kept for 30 min. The content of H$_2$O$_2$ was determined by absorbance at 350 nm with a UV-Vis spectrophotometer (The standard curve for H$_2$O$_2$ was shown in Supplementary Fig. 38). The effect of different catalyst concentrations on the photocatalytic H$_2$O$_2$ activity was also illustrated (Supplementary Fig. 39). More details of the process can be found in our previous paper[14]. The photocatalytic O$_2$ generation reaction was tested by connecting the reactor to a glass-enclosed gas system (Labsolar-6A, PerfectLight). Disperse 30 mg of the catalyst in 30 ml of pure water, and add AgNO$_3$ (20 mM) as an electron acceptor and 30 mg of La$_2$O$_3$ to stabilize the pH of the reaction system. N$_2$ was passed through the reactor for 30 min to remove residual gas. The generated O$_2$ was detected under visible light irradiation (λ ≥ 420 nm) of a 300 W Xe lamp and by an online gas chromatograph (5 Å molecular sieve column, Ar carrier).

### Determination of AQY efficiency

The AQY efficiency of Ni$_{SAPs}$-PuCN was tested under pure water (50 mg catalyst in 50 ml solution, pH = 7, 25 °C). AQY tests were performed using a 300 W Xe lamp (PLS-FX300HU, PerfectLight) with different bandpass filters at 400, 420, 450, 500 and 550 nm (FWHM = 15 nm). The average light intensity was measured by an optical power meter (Thorlabs) and the irradiated area was controlled at 1.69 cm$^2$. AQY is calculated by the following Eq. (1):

$$AQY = \frac{2 \times H_2O_2 \, \text{formed (mol)}}{\text{the number of incident photons (mol)}} \times 100\% \qquad (1)$$

### Measurement of SCC efficiency

The SCC efficiency was evaluated using a 300 W Xe lamp (PLS-FX300HU, PerfectLight) with an AM 1.5 G filter as a simulated sun light source. 500 mg of the catalyst was dispersed in 100 ml of pure water (pH = 7, 25 °C) with sufficient O$_2$ for photoreaction. The spot irradiation area is set to 1 cm$^2$, and the light intensity is strictly set to 100 mW cm$^{-2}$ (1 sun) by the optical power meter (Thorlabs). The SCC efficiency is calculated by the following Eq. (2):

$$SCC = \frac{\left[\Delta G_{H_2O_2}\right] \times \left[N_{H_2O_2}\right]}{I \times S \times T} \times 100\% \qquad (2)$$

where $\Delta GH_2O_2$ is the Gibbs free energy (117 KJ mol$^{-1}$) of forming H$_2$O$_2$, N$H_2O_2$ is the amount of H$_2$O$_2$ produced, I is the light intensity of simulated sunlight (100 mW cm$^{-2}$), S is the illuminated area (1 cm$^2$), and T is the illuminated time (s).

### Electrochemical measurement of O$_2$ reduction reaction (ORR)

The number of transferred electrons (n) and H$_2$O$_2$ selectivity of the samples in ORR were measured using a rotating ring disk electrode (RRDE). The electrochemical measurement adopts a three-electrode system, in which Pt/C is the counter electrode, RRDE is the working electrode, and Ag/AgCl is the reference electrode. The electrolyte is O$_2$ saturated 0.1 M KOH solution. The speed of RRDE was used at 1600 rpm and the potential range was set to 0−1.0 V vs. RHE. The catalyst ink on the RRDE working electrode is prepared as follows. After uniform grinding of 4 mg catalyst, it was ultrasonically dispersed in 400 μL pure water, 600 μL isopropanol and 10 μL Nafion solution for 1 h. Next, 10 μL of ink was dropped on the RRDE electrode and dried at room temperature. The number of transferred electrons is calculated according to the following Eq. (3):

$$n = \frac{4I_d}{I_d + I_r/N} \qquad (3)$$

The H$_2$O$_2$ selectivity is calculated according to the following Eq. (4):

$$H_2O_2(\%) = 2 \times \frac{I_r/N}{I_d + I_r/N} \times 100\% \qquad (4)$$

where I$_r$ is the ring current, I$_d$ is the disc current, and N is the collection efficiency (0.37).

### Characterization

TEM, HAADF-STEM and EDS were collected on a spherical aberration-corrected transmission electron microscope FEI Titan Themis with an accelerating voltage of 300 KV. The surface morphology of the samples was characterized by SEM (Thermo Fisher Scientific Quattro S). The X-ray diffraction (XRD) patterns of the samples were collected on an X-ray diffractometer (Bruker D8) with Cu Kα radiation at 40 kV and 40 mA. The Fourier transform infrared (FTIR) spectra were obtained on Bruker V70 spectrometer. The single-atom content in M$_{SAPs}$-PuCN was determined by inductively coupled plasma mass spectrometry (ICP-MS) on a PerkinElmer NexION 300X (samples were dissolved in aqua regia). X-ray photoelectron spectroscopy (XPS) measurements were acquired on an ESCALAB 250Xi instrument (Thermo Fisher Scientific) using Al Kα radiation, and the calibration peak was C 1s at 284.8 eV. UV-Vis diffuse reflectance spectroscopy (DRS) of the samples

was collected on Shimadzu UV-3600 instrument with BaSO$_4$ as reference. Photoluminescence (PL) spectra were measured on a fluorescence spectrometer (Hitachi F-7000) with an excitation wavelength of 325 nm. Time-resolved photoluminescence (TRPL) spectra were collected on an Edinburgh FLS1000 fluorescence spectrometer with excitation at 375 nm. Raman spectra were acquired on a high-resolution confocal Raman spectrometer (RENISHAW inVia) with an excitation laser of 785 nm. The EPR signal generated by the samples in the 5,5-dimethyl-1-pyrroline N-oxide solution (DMPO) was captured using an A300-10/12 spectrometer. The N$_2$ adsorption-desorption isotherm and pore size distribution of the samples were measured at 77 K using a micrometrics Max-II system. Detection of temperature programmed O$_2$ desorption (TPD-O$_2$) of samples using a temperature-programmed chemisorption instrument (AutoChem1 II 2920). The electrochemical impedance spectroscopy (EIS) was measured on the electrochemical workstation (CHI600A) with a standard three-electrode system, in which the catalyst-coated ITO was the working electrode, the Pt wire was the counter electrode and the saturated calomel electrode was the reference electrode.

## In situ XAFS measurement

The in situ XAFS measurements of Ni $K$-edge were carried out at the 1W1B station in the Beijing Synchrotron Radiation Facility (BSRF), China. The storage ring of BSRF was operated at 2.5 GeV with a maximum current of 250 mA. The beam from the bending magnet was monochromatized utilizing a Si (111) double-crystal monochromator and further detuning of 30% to remove higher harmonics. The photochemical in situ XAFS tests were performed by a home-made cell in a pure water solution. The XAFS spectra were collected through fluorescence mode. The Ni$_{SAPs}$-PuCN catalyst on 3D substrate was cut into 1 × 2 cm$^2$ pieces and then sealed in a cell by Kapton film. In order to obtain information on the evolution of active sites during photochemical reactions, a series of representative working conditions were applied to the samples, including Ar-saturated solution (In Ar), O$_2$ saturated solution (In O$_2$) and light conditions (visible light irradiation in O$_2$ saturated solution). A 300 W Xe lamp (PLS-SXE 300D/DUV, PerfectLight) with a 420 nm cut-off film (λ ≥ 420 nm) was utilized as the light source for the photocatalytic reaction. During the collection of XAFS measurements, the position of the absorption edge ($E_O$) was calibrated using a standard sample of Ni, and all XAFS data were collected during one period of beam time.

## XAFS data analysis

The acquired EXAFS data were processed according to the standard procedures using the ATHENA module implemented in the IFEFFIT software packages. Subsequently, $k^3$-weighted χ(k) data in the k-space ranging from 2.6–11.8 Å$^{-1}$ were Fourier transformed to real space using a Hanning window (dk = 1.0 Å$^{-1}$) to separate the EXAFS contributions from different coordination shells. The best background removal was at the $R_{bkg}$ = 1.0 Å, and the low-frequency noise was removed fully. As for the Ni$_{SAPs}$-PuCN under saturated Ar solution, the curve fitting was done on the $k^3$-weighted EXAFS function χ(k) data in the k-range of 2.6–11.8 Å$^{-1}$ and in the R-range of 1.0–2.2 Å. The number of independent points for these samples are $N_{ipt}$ = 2Δk·ΔR/π = 2 × (11.8 − 2.6) × (2.2 − 1.0)/π = 7. However, the first coordination peak of Ni$_{SAPs}$−PuCN under O$_2$ saturated solution conditions showed similar intensity and higher R shift compared with Ar-saturated solution, which was ascribed to the fracture of Ni-N bond and addition of Ni−O coordination. Therefore, the two subshells of Ni−N and Ni−O coordination were considered for the curve fitting of Ni$_{SAPs}$−PuCN under O$_2$ saturated solution. During curve fittings, each of the Debye−Waller factors ($σ^2$), coordination numbers ($N$), interatomic distances ($R$), and energy shift ($ΔE_O$) was treated as adjustable parameters.

## Computational methods

All density functional theory calculations were performed using the Perdew-Burke-Ernzerhof formulation within the generalized gradient approximation. These calculations were conducted with the Vienna Ab Initio Package[53,54]. The projected augmented wave potential was chosen to describe the ion nucleus[55]. The valence electrons were considered, using a plane wave basis set with a kinetic energy cutoff of 500 eV. Electron energies are considered self-consistent when the energy change is less than 10$^{-5}$ eV. The geometry optimization was considered converged when the force change was smaller than 0.02 eV/Å. Grimme's DFT-D3 method is used to account for dispersion interactions[56].

The lattice constants of corrugated 2 × 2 g-C$_3$N$_4$ monolayer in a vacuum with a depth of 20 Å was optimized, when using a 2 × 2 × 1 k-point grid for Brillouin zone sampling, to be $a$ = 13.674 Å. This model comprises of 24 C and 32 N atoms. This model and metal single-atom-doped ones were used for adsorption. The adsorption energy ($E_{ads}$) of adsorbate A was defined as: $E_{ads} = E_{A/surf} − E_{surf} − E_A(g)$, where $E_{A/surf}$ represents the energy of the adsorbate A adsorbed on the surface, $E_{surf}$ and $E_A(g)$ is the energy of isolated A molecule in a cubic periodic box with a side length of 20 Å and a 1 × 1 × 1 Monkhorst-Pack k-point grid for Brillouin zone sampling, respectively. The free energy of gas-phase molecules or surface adsorbates is calculated from the equation G = E + ZPE − TS, where E is the total energy, ZPE is the zero-point energy, T is the temperature in Kelvin (298.15 K is set here), and S is the entropy. The overall reaction pathway for the calculated 2e$^-$ ORR includes: (1) O$_2$ adsorption on the active site (*) in catalysts (Eq. 5); (2) the adsorbed O$_2$ captures an e$^-$ and combines with H$^+$ to form *OOH (Eq. 6); (3) *OOH continues the electron-coupled proton transfer reaction to form H$_2$O$_2$ (Eq. 7); (4) desorption and diffusion of H$_2$O$_2$ (Eq. 8).

$$O_2(g) + * \rightarrow *O_2 \qquad (5)$$

$$*O_2 + e^- + H^+ \rightarrow *OOH \qquad (6)$$

$$*OOH + e^- + H^+ \rightarrow *H_2O_2 \qquad (7)$$

$$*H_2O_2 \rightarrow * + H_2O_2(l) \qquad (8)$$

## Data availability

The data that support the findings of this study are available within the article and the Supplementary Information. The source data are available from the corresponding authors upon request.

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

## Acknowledgements

This work was supported by the National Key Projects for Fundamental Research and Development of China (2021YFA1500803, T.Z.), National Natural Science Foundation of China (12074015 (K.Z.), 51825205 (T.Z.), 22241202 (Q.L.)), the Beijing Outstanding Young Scientists Projects (BJJWZYJH01201910005018, K.Z.), the CAS Project for Young Scientists in Basic Research (YSBR-004, T.Z.), and National Youth Fund (22108093, Y.C.).

## Author contributions

X.Z. synthesized and characterized the samples, performed experiments and theoretical calculations, and contributed to the concept of this research. X.Z. and H.S. conceptualized the project. H.S., P.C., and Q.L. provided guidance on XAFS testing and analysis. L.G., Y.C., and Y.W. assisted and guided the theoretical calculations. Y.F., R.F., J.H., X.Z., P.M., H.H., and K.W. assisted in characterization of the samples. Y.Z., C.W., Z.T., and Q.Z. provided constructive suggestions. X.Z., H.S., Q.L., T.Z., and K.Z. analysed the experimental results and wrote the manuscript. K.Z. led the entire project. All authors read the manuscript and contributed to the discussion of the results.

## Competing interests

The authors declare no competing interests.
