## [Peer Review File · Nature Communications]

Developing Ni single-atom sites in carbon nitride for efficient photocatalytic H₂O₂ productionREVIEWER COMMENTS

Reviewer #1 (Remarks to the Author):

In this manuscript, Zhang and co-authors developed high-density Ni single-atom CN-based photocatalysts for H₂O₂ generation, and investigated the structure evolution mechanism of Ni single-atom sites in surface reactions. The NiSAPs-PuCN in the manuscript exhibited excellent H₂O₂ generation activity in pure water (SCC of 0.82%, AQY:10.9% at 420 nm), which is superior to previous reports of this material in the same system. The authors used in situ X-ray absorption spectroscopy to reveal the structural evolution of Ni single-atom sites, and proved that this structural evolution is closely related to its high activity. The atomic-level Ni active site evolution mechanism proposed by the authors is novel and well-documented by experiments and theoretical calculations. I think the material design, high-performance realization and research ideas on active sites in the manuscript are innovative in the photocatalytic system.

The experimental methods used for the analysis of the atomistic and electronic structure represent the state of the art in the field and are very well suited to obtain the required information. The theoretical approach used to corroborate the experimental findings and to assess the generic properties is standard, but suitable for the purpose. The procedure is concise and transparent, and the results are convincing and fully supporting the conclusions drawn. The paper is original and well written, addresses an interesting and timely topic, and should be of interest to a larger readership.

However, the authors should be address the following issues to improve the quality of this work:

λ Increasing the metal single-atom loading plays an important role in the further development and application of single-atom catalysts. The authors proposed a universal synthesis method for high-load single-atom (~10 wt%) photocatalysts, but the introduction of current literature on high-density single-atom photocatalysts is missing in the description part of the manuscript. The authors are suggested to introduce these literatures (nature synthesis: 10.1038/s44160-022-00129-x; nature nanotechnology:10.1038/s41565-022-01090-8) at appropriate places to illustrate the importance of highly loaded single-atom catalysts.

λ In the synthesis mechanism of H₂O₂, the authors need to clarify whether the hydroxyl radical ($\cdot\text{OH}$) is involved in the formation reaction of H₂O₂. Relevant experiments need to be supplemented for clarification.

λ In the in situ XAFS measurement of Figure 5a, the authors performed the reaction under O₂ atmosphere and light conditions. The authors should state the type of light source used for the lighting conditions.

λ The authors demonstrate that high-density Ni single atoms have better catalytic activity than Ni nanoclusters. I think the experimental samples should be supplemented to compare and analyze the effects of Ni single atoms, Ni nanoclusters and Ni nanoparticles on the H₂O₂ production activity.

λ In line 325 of the manuscript, "Compared with BCN, the absorbed O₂ on NiSAPs-PuCN has longer O=O bond length (1.29 Å)". The authors did not describe the bond length of the adsorbed O₂ on BCN.

λ The authors propose that the Ni single atom coordination structure in the sample is Ni-N₃. Is there a possibility of Ni-N₄ or Ni-N₆ coordination?

λ The authors have demonstrated that the dynamic structure evolution of Ni single atoms in 2e-ORR is the main reason for the high activity and selectivity of the catalyst. It should be noted that the Ni-N₃ site is transformed into O1-Ni-N₂ after O₂ adsorption. However, in Fig. 5c, the authors mark O1-Ni-N₂ as O-O-Ni-N₂, while in Fig. 5e it is expressed as O=O-Ni-N₂. The author should unify the expression form of O1-Ni-N₂.

λ The authors are suggested to supplement the photocurrent response measurements and correlation analysis of the samples to illustrate the effect of Ni single atoms on the carrier separation and transport.

λ On page 13, line 241, there is a wrong sentence: which was confirmed by a series of spectroscopic measurements.

λ On page 21, line 398, there is a formatting error: 550°C. There should be a space between the number and the unit.

Reviewer #2 (Remarks to the Author):

This paper reports the preparation and mechanism study of Ni single-atom carbon nitrogen materials with porous structure for the photocatalytic synthesis of H₂O₂. High content of Ni single-atom samples achieves high H₂O₂ production efficiency in pure water. The authors detail the effects of Ni single atoms on the electronic structure, optoelectronic properties, and reaction mechanism of CN. By using in situ EXAFS and Raman, they propose a structural change of Ni single-atom sites during the H₂O₂ reaction: Ni-N₃ to O₁-Ni-N₂ to OOH-Ni-N₂. Their theoretical calculations and auxiliary experiments show that Ni site evolution effectively activates O₂ to enhance the catalytic activity and selectivity. Investigating the relationship between the structural changes of single-atom sites during the reaction and the high activity is the highlight of this paper. This work is carefully done and provides reference value to the photocatalytic H₂O₂ synthesis. I recommend this work for publication with minor modifications. The comments and concerns are listed as follows.

1. The authors chose a catalyst concentration of 1 g/L for photocatalytic H₂O₂ activity measurements (mol/g/h). Although many researchers tend to use mol/g/h to compare the photocatalytic activity, it is necessary to provide the influence of different catalyst dosage on the catalytic activity.
2. The authors propose a synthetic strategy for high-density single-atom catalysts based on carbon nitride. I wonder if this method can be applied to other photocatalytic materials? Such as TiO₂?
3. In theoretical calculations, the authors analyzed in detail the effect of O₂ adsorption on Ni samples on the activity and selectivity of H₂O₂. On the BCN model, although the adsorption and activation of O₂ by BCN is weak, the authors should explain that O₂ tends to adsorb on C atoms, N atoms or other positions.
4. According to the AFM measurement (Supplementary Figure 11), the thickness of NiSAPs-PuCN is about 2~4 nm, so how many layers it contains? The g-C₃N₄ is a typical 2D layered material, and this physical characteristic of the catalyst should be described.
5. In Supplementary Figure 16, the authors pointed out that high-loaded NiSAPs-PuCN had higher catalytic activity than other low-loaded samples (NiSAPs-PuCN1-3). Only HAADF-STEM and ICP showed that the Ni single-atom state/loading content in NiSAPs-PuCN1-3 is not comprehensive, at least the overall morphology and EDS of these samples are needed to reflect specific information.
6. The authors have used "O=O" extensively to indicate the bonding of O₂ molecules in the manuscript. In line 298, the authors express O₂ as "O-O". The same problem occurs in Figure 6b. Please revise.
7. In the TPD analysis, the authors concluded that NiSAPs-PuCN has stronger O₂ surface binding strength than BCN because it has remarkable stronger O₂ adsorption capacity. It should be noted that the adsorption capacity is related to its surface site while the binding strength should be derived from the desorption temperature.
8. The authors need to explain the meaning of the yellow/blue regions expressed in the charge difference density in Figure 6g-h.

Reviewer #3 (Remarks to the Author):

The authors report a study for the photocatalytic hydrogen peroxide (H₂O₂) production using N sites embedded into carbon nitride hosts. The derivation of porous ultrathin carbon nitrides from bulky g-CN structures (BCN) is interesting in that they enable the high-mass loading of Ni sites. However, thermal stripping and exfoliation to derive porous structures appear to be on the very well-known

conventional processes. Besides, the demonstration of single metal atoms should be elaborate. Moreover, the authors need to demonstrate that single atoms are maintained after the repeated photocatalytic cycles as well as long-hour operation.

Also, the quantum yield at 420 nm that is not the major spectrum of sun light, is considered to be not making a great breakthrough. the solar-to-chemical conversion conversion (0.82%) is not much superior to the previous results (Figure 4g). Besides, the quantum yields in the visible-light spectra over 450 nm are shown to be significantly degrading, as shown in Figure 4c.

Besides, the simultaneous mechanisms for exciton (electron-hole pair) transfer involving in the photocatalytic reduction of O₂ into H₂O₂ are unclear.

Reply letter for manuscript: Revealing structure evolution mechanism of Ni single-atom sites in carbon nitride for efficient photocatalytic H₂O₂ production (NCOMMS-23-18755A).

Reviewer #1 (Remarks to the Author):

In this manuscript, Zhang and co-authors developed high-density Ni single-atom CN-based photocatalysts for H₂O₂ generation, and investigated the structure evolution mechanism of Ni single-atom sites in surface reactions. The Ni_{SAPs}-PuCN in the manuscript exhibited excellent H₂O₂ generation activity in pure water (SCC of 0.82%, AQY:10.9% at 420 nm), which is superior to previous reports of this material in the same system. The authors used in situ X-ray absorption spectroscopy to reveal the structural evolution of Ni single-atom sites, and proved that this structural evolution is closely related to its high activity. The atomic-level Ni active site evolution mechanism proposed by the authors is novel and well-documented by experiments and theoretical calculations. I think the material design, high-performance realization and research ideas on active sites in the manuscript are innovative in the photocatalytic system.

The experimental methods used for the analysis of the atomistic and electronic structure represent the state of the art in the field and are very well suited to obtain the required information. The theoretical approach used to corroborate the experimental findings and to assess the generic properties is standard, but suitable for the purpose. The procedure is concise and transparent, and the results are convincing and fully supporting the conclusions drawn. The paper is original and well written, addresses an interesting and timely topic, and should be of interest to a larger readership.

However, the authors should be address the following issues to improve the quality of this work.

Reply:

We sincerely thank the reviewer for taking the time to review the manuscript and giving positive comments. We are very grateful to this reviewer for his/her recognition and high evaluation of the high H₂O₂ generation catalytic activity and atomic-level structure evolution mechanism of Ni_{SACs}-PuCN in this work. This reviewer's professional, academic comments really help us to further improve the quality of the work. We respond to the reviewer's comments by adding more comprehensive experiments and theoretical calculations. Accordingly, based on these constructive comments and suggestions, we have made corresponding revisions in the manuscript and highlighted the revised parts with a bright yellow background for easy tracking. With these improvements, we believe that the revised manuscript can better meet the high standards of *Nature Communications*. **The point-by-point responses are as follows:**

Reviewer #1-Comment 1:

Increasing the metal single-atom loading plays an important role in the further development and application of single-atom catalysts. The authors proposed a universal synthesis method for high-load single-atom (~10 wt%) photocatalysts, but the introduction of current literature on high-density single-atom photocatalysts is missing in the description part of the manuscript.

The authors are suggested to introduce these literatures (nature synthesis: 10.1038/s44160-022-00129-x; nature nanotechnology:10.1038/s41565-022-01090-8) at appropriate places to illustrate the importance of highly loaded single-atom catalysts.

Reply: Thanks for constructive suggestions. We have cited the above references where appropriate in the manuscript. In the revised manuscript, we have made the following changes: **Increasing the single atom loading to create more active sites is beneficial to improve the catalytic activity^{39,40}.**

Reviewer #1-Comment 2:

In the synthesis mechanism of H₂O₂, the authors need to clarify whether the hydroxyl radical ($\cdot\text{OH}$) is involved in the formation reaction of H₂O₂. Relevant experiments need to be supplemented for clarification.

Reply: Thanks for the high valuable comment. **We further verified that $\cdot\text{OH}$ radicals will not participate in the generation of H₂O₂ by supplementary radical trapping experiments.** As shown in **Figure 1 below**, related experiment and discussions were supplemented in the revised **Supplementary Information (Supplementary Figure 19)**.

Figure 1 (Supplementary Figure 19 of revised Supporting Information). The photocatalytic H₂O₂ generation rates of Ni_{SAPs}-PuCN under different reaction gases or different sacrificial agents.

In **Supplementary Figure 19**, when the O₂ in the reaction was replaced by N₂, almost no H₂O₂ was detected, indicating that O₂ is a necessary reactant. To further understand the photocatalytic H₂O₂ evolution mechanism over Ni_{SAPs}-PuCN in pure water, AgNO₃ and benzoquinone (BQ) were used as electron (e⁻) and superoxide radical scavengers to perform active species trapping experiments. When AgNO₃ (0.1 mM) was added to the reaction system, the H₂O₂ yield dropped rapidly, which indicated that H₂O₂ was generated by electron reduction of O₂. Notably, when BQ (0.1 mM) was added to the system, the yield of H₂O₂ was very low, suggesting that $\cdot\text{O}_2^-$ is a necessary intermediate for the generation of H₂O₂ ($\text{O}_2 \rightarrow \cdot\text{O}_2^- \rightarrow \text{H}_2\text{O}_2$). **The reaction mechanism was explored using tert-butanol (TBA) as a hydroxyl radical ($\cdot\text{OH}$) scavenger. After adding TBA, the H₂O₂ activity of Ni_{SAPs}-PuCN remained basically unchanged, which indicated that hydroxyl radicals did not participate in the generation of H₂O₂.**

Reviewer #1-Comment 3:

In the in situ XAFS measurement of Figure 5a, the authors performed the reaction under O₂ atmosphere and light conditions. The authors should state the type of light source used for the lighting conditions.

Reply: We thank the reviewer for the constructive comments. We have added this experimental detail in the revised manuscript. Corresponding changes were made in the “Methods” section (In situ XAFS measurements) of the revised manuscript as follows:

A 300 W Xe lamp (PLS-SXE 300D/DUV, PerfectLight) with a 420 nm cut-off film ($\lambda \geq 420$ nm) was utilized as the light source for the photocatalytic reaction.

Reviewer #1-Comment 4:

The authors demonstrate that high-density Ni single atoms have better catalytic activity than Ni nanoclusters. I think the experimental samples should be supplemented to compare and analyze the effects of Ni single atoms, Ni nanoclusters and Ni nanoparticles on the H₂O₂ production activity.

Reply: Thanks to the reviewer for the valuable information. According to the reviewer's constructive comments, we synthesized Ni nanoparticles on PuCN (named Ni Nps-PuCN), and performed characterization and photocatalytic H₂O₂ activity tests. The activity comparison with Ni single-atom samples (Ni_{SAPs}-PuCN) under the same conditions showed that the catalytic activity of Ni single atoms on PuCN was much higher than that of Ni nanoparticles, which further demonstrated the advantages of Ni single atoms in the manuscript. As shown in **Figure 2 below**, supplementary experiments and discussion were collated in the revised **Supplementary Information (Supplementary Figure 18)**.

Figure 2 (Supplementary Figure 18). (a) HAADF-STEM image and (b-d) EDS mapping of Ni Nps-PuCN. (e) Comparison of the photocatalytic H₂O₂ activity of BCN, PuCN, Ni_{SAPs}-PuCN, and Ni Nps-PuCN (Pure water, $\lambda \geq 420$ nm, 60 mW cm⁻²; 30 mg catalyst in 30 ml pure water, 1 g L⁻¹ catalyst; 25 °C).

We loaded Ni nanoparticles on PuCN (named Ni Nps-PuCN) using the NaBH₄ reduction method. As shown in **Supplementary Figure 18a**, there are abundant Ni nanoparticles on Ni

Nps-PuCN, indicating that Ni nanoparticles have been successfully loaded on PuCN. The EDS images (**Supplementary Figure 18b-d**) further show that C and N are uniformly distributed on the Ni Nps-PuCN, and Ni elements are concentrated on the nanoparticles. The photocatalytic H₂O₂ activities of Ni_{SAPs}-PuCN and Ni Nps-PuCN were measured under the same conditions. As shown in Figure **Supplementary Figure 18e**, the H₂O₂ activity of Ni_{SAPs}-PuCN is significantly higher than that of Ni Nps-PuCN, which indicates that Ni single atoms on PuCN have stronger catalytic activity than Ni nanoparticles.

In response to this issue, in the main text of the revised manuscript, we make the following changes:

Meanwhile, the effect of Ni single atoms loading content and Ni nanoparticles on the H₂O₂ activity was further sorted out (**Supplementary Figs. 17 and 18**).

Reviewer #1-Comment 5:

In line 325 of the manuscript, "Compared with BCN, the adsorbed O₂ on Ni_{SAPs}-PuCN has longer O=O bond length (1.29 Å)". The authors did not describe the bond length of the adsorbed O₂ on BCN.

Reply: Thanks for your careful review. We have added this information in the revised manuscript as follows:

Compared with BCN (the bond length of adsorbed O₂ on BCN is 1.23 Å), the adsorbed O₂ on Ni_{SAPs}-PuCN has longer O=O bond length (1.29 Å) and larger charge transfer, indicating that the O₁-Ni-N₂ intermediate structure promotes the O₂ adsorption and activation.

Reviewer #1-Comment 6:

The authors propose that the Ni single atom coordination structure in the sample is Ni-N₃. Is there a possibility of Ni-N₄ or Ni-N₆ coordination?

Reply: Thanks for the reviewer's comment. In the previous manuscript, it was determined that Ni single atoms in Ni_{SAPs}-PuCN are coordinated by Ni-N₃ through synchrotron radiation fitting (**Figure 3c**) and theoretical calculation (**Supplementary Figure 15**). The detailed analysis is shown in **Figure 3**:

Figure 3 (Supplementary Figure 16 in the revised Supplementary Information). Initial and optimized coordination structures of Ni single atoms on g-C₃N₄ structural units (Light green: Ni single atom. Blue: N atoms. Grey: C atoms).

The coordination information of Ni single atoms in g-C₃N₄ was further verified using theoretical calculations. As shown in **Supplementary Figure 16**, the Ni-N₃ coordination can be stably formed after optimization. On one hand, the theoretically calculated coordination number of Ni single atom is consistent with EXAFS. On the other hand, the average Ni-N bond length provided by theoretical calculation is 1.98 Å, which matches the average Ni-N bond length (2.07 Å) measured by XAFS as high as 96%.

At the same time, we performed additional theoretical calculations (as shown in Figure 4 below), and the results show that it is difficult to form Ni-N₄ and Ni-N₆ coordination in g-C₃N₄ structure.

Figure 4. Initial coordination (Ni-N₄ and Ni-N₆) and optimized coordination structure of Ni single atom on g-C₃N₄ structural unit (Light green: Ni single atom. Blue: N atoms. Grey: C atoms).

As shown in **Figure 4** above, we set Ni single atom to coordinate with Ni-N₄ and Ni-N₆ in the g-C₃N₄ structure, and the calculation results after optimization all showed Ni-N₃ coordination. Based on the above analysis, it is difficult for g-C₃N₄ structure to have Ni-N₄ and Ni-N₆ coordination, which fully justifies the rationality of Ni-N₃ coordination in the manuscript.

Reviewer #1-Comment 7:

The authors have demonstrated that the dynamic structure evolution of Ni single atoms in 2e⁻ ORR is the main reason for the high activity and selectivity of the catalyst. It should be noted that the Ni-N₃ site is transformed into O₁-Ni-N₂ after O₂ adsorption. However, in Fig. 5c, the authors mark O₁-Ni-N₂ as O-O-Ni-N₂, while in Fig. 5e it is expressed as O=O-Ni-N₂. The author should unify the expression form of O₁-Ni-N₂.

Reply: Thanks again to the reviewer for the recognition and careful review of the evolution mechanism of Ni single-atom active sites (Ni-N₃ → O₁-Ni-N₂ → OOH-Ni-N₂). According to

the reviewer's suggestion, we collectively named the O_1 -Ni- N_2 sites formed after the Ni- N_3 sites adsorbed O_2 in Figure 5e-f of the revised manuscript as $O=O$ -Ni- N_2 . The revised Figure 5 in the manuscript is shown in **Figure 5** below.

Figure 5 (Figure 5 of revised manuscript).

Reviewer #1-Comment 8:

The authors are suggested to supplement the photocurrent response measurements and correlation analysis of the samples to illustrate the effect of Ni single atoms on the carrier separation and transport.

Reply: Thanks for your comments. We supplemented the transient photocurrent response measurements of the samples and organized the results in the revised Supplementary Information (**Supplementary Figure 27d**), as shown in **Figure 6** below.

Figure 6 (Supplementary Figure 27). (a) Photoluminescence (PL) spectra, (b) electrochemical chemical impedance spectra (EIS), (c) time-resolved photoluminescence (TRPL) spectra, and (d) transient photocurrent response (TPR) of BCN, PuCN, and NiSAPs-PuCN.

Additional discussion follows:

In Supplementary Fig. 27d, compared to BCN and PuCN, NiSAPs-PuCN exhibited a stronger photocurrent density after turning on the light, indicating that the introduction of Ni single atoms facilitated the transport of photogenerated carriers.

Reviewer #1-Comment 9:

On page 13, line 241, there is a wrong sentence: which was confirmed by a series of spectroscopic measurements.

Reply: Thanks for your careful review. In the revised manuscript we have made changes:

The introduction of Ni single atoms effectively promotes carrier separation and transport, as confirmed by a series of spectroscopic measurements (Supplementary Fig. 27).

Reviewer #1-Comment 10:

On page 21, line 398, there is a formatting error: 550°C. There should be a space between the number and the unit.

Reply: Thanks again for your careful review. This was caused by our carelessness, and we are sorry for that. In the revised manuscript, we have corrected this:

Similar to the synthesis of BCN, the holding time at 550 °C in the muffle furnace was extended to 4 h.

Reviewer #2 (Remarks to the Author):

This paper reports the preparation and mechanism study of Ni single-atom carbon nitrogen materials with porous structure for the photocatalytic synthesis of H₂O₂. High content of Ni single-atom samples achieves high H₂O₂ production efficiency in pure water. The authors detail the effects of Ni single atoms on the electronic structure, optoelectronic properties, and reaction mechanism of CN. By using in situ EXAFS and Raman, they propose a structural change of Ni single-atom sites during the H₂O₂ reaction: Ni-N₃ to O₁-Ni-N₂ to OOH-Ni-N₂. Their theoretical calculations and auxiliary experiments show that Ni site evolution effectively activates O₂ to enhance the catalytic activity and selectivity. Investigating the relationship between the structural changes of single-atom sites during the reaction and the high activity is the highlight of this paper. This work is carefully done and provides reference value to the photocatalytic H₂O₂ synthesis. I recommend this work for publication with minor modifications. The comments and concerns are listed as follows.

Reply:

First of all, we are grateful to this reviewer for the innovative recognition and constructive comments on our manuscript. Meanwhile, we thank this reviewer for his/her affirmation and positive comments on the high H₂O₂ activity and catalytic reaction mechanism in this work. The reviewer's constructive comments are helpful to further improve the quality of our manuscript. We respond to the reviewer's high-quality comments by adding more comprehensive experiments and theoretical calculations. Based on these constructive comments and suggestions, we have made corresponding revisions in the manuscript and highlighted the revised parts with a bright yellow background for easy tracking. With these improvements, we believe that the revised manuscript can better meet the high standards of *Nature Communications*. **The point-by-point responses are as follows:**

Reviewer #2-Comment 1:

The authors chose a catalyst concentration of 1 g/L for photocatalytic H₂O₂ activity measurements (mol/g/h). Although many researchers tend to use mol/g/h to compare the photocatalytic activity, it is necessary to provide the influence of different catalyst dosage on the catalytic activity.

Reply: Thanks to the reviewer for the valuable information. Following the reviewer's suggestion, we added the effect of different catalyst concentrations (Ni_{SAPs}-PuCN) on the photocatalytic H₂O₂ activity in the Supplementary Information (**Supplementary Figure 39**), illustrating that 1 g L⁻¹ catalyst concentration was the optimal catalyst dosage for this work.

We added relevant descriptions in the “Methods” section of the manuscript: **The effect of different catalyst concentrations on the photocatalytic H₂O₂ activity was also illustrated (Supplementary Fig. 39).**

The Supplementary experimental results are shown in Figure 7 below.

Figure 7 (Supplementary Figure 39). The effect of different catalyst concentrations of NiSAPS-PuCN on the photocatalytic H₂O₂ activity in this work ($\lambda \geq 420$ nm, pure water) ($\lambda \geq 420$ nm, 60 mW cm⁻²; 30 ml pure water; 25 °C).

As shown in **Supplementary Figure 39**, the catalyst concentration to measure the photocatalytic H₂O₂ generation rate ($\mu\text{mol g}^{-1} \text{h}^{-1}$) was confirmed to be 1 g L⁻¹ (30 mg in 30 mL) in this work.

Reviewer #2-Comment 2:

The authors propose a synthetic strategy for high-density single-atom catalysts based on carbon nitride. I wonder if this method can be applied to other photocatalytic materials? Such as TiO₂?

Reply: Thank you for acknowledging the strategy for the synthesis of high-loaded single-atom catalysts in this manuscript. **Adjusting the microscopic morphology of the catalyst and further combining continuous ultrasonic wet chemical precipitation is beneficial to the high-density and uniform loading of single atoms on the catalyst.** Using this strategy, we also loaded high-density Pt single atoms on TiO₂ (Pt_{SACS}-TiO₂), and characterized the samples using aberration-corrected transmission electron microscopy, as shown in **Figure 8** below.

Figure 8. (a-d) EDS mapping images of Pt_{SACS}-TiO₂. (e) Aberration-corrected HAADF-STEM image of Pt_{SACS}-TiO₂.

As shown in **Figure 8a-d**, the EDS images show that Ti, O and Pt elements are uniformly distributed on the TiO₂ substrate. And **Figure 8e** further demonstrates that the high-density Pt single atoms are uniformly anchored on TiO₂.

Reviewer #2-Comment 3:

In theoretical calculations, the authors analyzed in detail the effect of O₂ adsorption on Ni samples on the activity and selectivity of H₂O₂. On the BCN model, although the adsorption and activation of O₂ by BCN is weak, the authors should explain that O₂ tends to adsorb on C atoms, N atoms or other positions.

Reply: Thank you for your nice question. We added the discussion of the details of O₂ adsorption on BCN through theoretical calculations (the supplementary content was placed in the revised **Supplementary Figure 36**), as shown in **Figure 9** below.

	O ₂ adsorption on BCN	
	Situation 1	Situation 2
Before optimization		After optimization		O ₂ adsorption energy	1.12 eV	1.10 eV

Figure 9 (Supplementary Figure 36). The adsorption of O₂ on BCN (before and after optimization) was investigated by theoretical calculations.

As shown in **Supplementary Figure 36**, two possible adsorption situations of O₂ on BCN were considered, and the optimized results of theoretical calculations showed the preferred adsorption sites of O₂ on carbon nitride (near N sites). At the same time, the adsorption energy of O₂ in the two cases is positive and the difference is small (1.12 eV and 1.10 eV), indicating that BCN has a weak adsorption on O₂.

Reviewer #2-Comment 4:

According to the AFM measurement (Supplementary Figure 11), the thickness of Ni_{SAPs}-PuCN is about 2~4 nm, so how many layers it contains? The g-C₃N₄ is a typical 2D layered material, and this physical characteristic of the catalyst should be described.

Reply: Thanks for your constructive comments. A g-C₃N₄ material with a thickness of 1 nm contains approximately 3 atomic layers (Nature Energy, 10.1038/s41560-021-00795-9). The AFM measurement results in the manuscript show that the Ni_{SAPs}-PuCN has a thickness of 2 ~ 4 nm and contains about 6 ~ 12 g-C₃N₄ atomic layers. The content after modification is shown in the **Figure 10** below.

Figure 10 (Supplementary Figure 11). (a) AFM image of Ni_{SAPs}-PuCN and (b) the height profile determined along the line in the AFM image.

In **Supplementary Figure 11**, the AFM image also shows that Ni_{SAPs}-PuCN has a porous thin-layer structure, and the thickness of these undulating thin layers is only about 2 ~ 4 nm (about 6 ~ 12 atomic layers).

Reviewer #2-Comment 5:

In Supplementary Figure 16, the authors pointed out that high-loaded Ni_{SAPs}-PuCN had higher catalytic activity than other low-loaded samples (Ni_{SAPs}-PuCN₁₋₃). Only HAADF-STEM and ICP showed that the Ni single-atom state/loading content in Ni_{SAPs}-PuCN₁₋₃ is not comprehensive, at least the overall morphology and EDS of these samples are needed to reflect specific information.

Reply: Thank you for your constructive suggestion and valuable information. We supplemented the aberration-corrected electron microscopy characterization of the samples (Ni_{SAPs}-PuCN-1, Ni_{SAPs}-PuCN-2, Ni_{SAPs}-PuCN-4), which were further organized in the revised Supplementary Information (**Supplementary Figure 17f-h**), as shown in **Figure 11** below.

Figure 11 (Supplementary Figure 17). (a) Comparison of the photocatalytic H_2O_2 activity of $\text{Ni}_{\text{SAPs}}\text{-PuCN}$ with different Ni single atom loadings (Pure water, $\lambda \geq 420$ nm, 60 mW cm^{-2} ; 30 mg catalyst in 30 ml pure water, 1 g L^{-1} catalyst; $25 \text{ }^\circ\text{C}$). (b-e) HAADF-STEM images of different Ni single atom loadings (samples are $\text{Ni}_{\text{SAPs}}\text{-PuCN-1}$, $\text{Ni}_{\text{SAPs}}\text{-PuCN-2}$, $\text{Ni}_{\text{SAPs}}\text{-PuCN}$, $\text{Ni}_{\text{SAPs}}\text{-PuCN-4}$ in turn). The EDS mapping images of (f) $\text{Ni}_{\text{SAPs}}\text{-PuCN-1}$, (g) $\text{Ni}_{\text{SAPs}}\text{-PuCN-2}$, and (h) $\text{Ni}_{\text{SAPs}}\text{-PuCN-4}$.

The supplementary relevant description is:

In Supplementary Figure 17f-h, the EDS images of the samples ($\text{Ni}_{\text{SAPs}}\text{-PuCN-1}$, $\text{Ni}_{\text{SAPs}}\text{-PuCN-2}$, and $\text{Ni}_{\text{SAPs}}\text{-PuCN-4}$) with different Ni single-atom loadings demonstrate the distribution of Ni elements on the porous thin-layer $\text{g-C}_3\text{N}_4$.

Reviewer #2-Comment 6:

The authors have used "O=O" extensively to indicate the bonding of O₂ molecules in the manuscript. In line 298, the authors express O₂ as "O-O". The same problem occurs in Figure 6b. Please revise.

Reply: Thank you for your careful review.

We corrected the above problems in the manuscript, and uniformly used "O=O" to represent O₂.

Reviewer #2-Comment 7:

In the TPD analysis, the authors concluded that Ni_{SAPs}-PuCN has stronger O₂ surface binding strength than BCN because it has remarkable stronger O₂ adsorption capacity. It should be noted that the adsorption capacity is related to its surface site while the binding strength should be derived from the desorption temperature.

Reply: Thanks for your valuable suggestions. Based on the reviewer's comments, we have succinctly expressed the description of O₂-TPD and revised it in the manuscript as follows:

Furthermore, temperature-programmed desorption of O₂ (TPD-O₂, **Fig. 6c**) suggested that Ni_{SAPs}-PuCN has remarkable stronger O₂ adsorption capacity than BCN, which fully confirms our calculation results.

Reviewer #2-Comment 8:

The authors need to explain the meaning of the yellow/blue regions expressed in the charge difference density in Figure 6g-h.

Reply: Thanks for your comments and valuable suggestions. In the note to Figure 6 of the revised manuscript, we have added the relevant description:

Fig. 6g-h, Optimized models and charge difference density after BCN (g) and Ni_{SAPs}-PuCN (h) generate *OOH (The isosurface value is 0.0016 eV Å⁻³. Electron accumulation and consumption are indicated in yellow and blue, respectively).

Reviewer #3 (Remarks to the Author):

The authors report a study for the photocatalytic hydrogen peroxide (H_2O_2) production using N sites embedded into carbon nitride hosts. The derivation of porous ultrathin carbon nitrides from bulky g-CN structures (BCN) is interesting in that they enable the high-mass loading of Ni sites. However, thermal stripping and exfoliation to derive porous structures appear to be on the very well-known conventional processes. Besides, the demonstration of single metal atoms should be elaborate. Moreover, the authors need to demonstrate that single atoms are maintained after the repeated photocatalytic cycles as well as long-hour operation.

Also, the quantum yield at 420 nm that is not the major spectrum of sun light, is considered to be not making a great breakthrough. The solar-to-chemical conversion conversion (0.82%) is not much superior to the previous results (Figure 4g). Besides, the quantum yields in the visible-light spectra over 450 nm are shown to be significantly degrading, as shown in Figure 4c.

Besides, the simultaneous mechanisms for exciton (electron-hole pair) transfer involving in the photocatalytic reduction of O_2 into H_2O_2 are unclear.

Reply: We sincerely appreciate your comments and valuable suggestions on the manuscript. These constructive and academic comments help us to further improve the quality of the manuscript. We respond to the reviewer's high-quality comments by adding more comprehensive experiments and more nuanced explanations. Meanwhile, we further carefully explained the current evaluation mechanism of photocatalytic H_2O_2 activity and the excellent photocatalytic H_2O_2 activity of $\text{Ni}_{\text{SAPs}}\text{-PuCN}$ to the reviewer. Based on these constructive comments and suggestions, we have revised the manuscript accordingly and highlighted the revised parts with a bright yellow background for easy tracking. With these improvements, we believe that the revised manuscript can better meet the high standards of *Nature Communications*. **The point-by-point responses are as follows:**

Reviewer #3-Comment 1:

The authors report a study for the photocatalytic hydrogen peroxide (H_2O_2) production using N sites embedded into carbon nitride hosts. The derivation of porous ultrathin carbon nitrides from bulky g-CN structures (BCN) is interesting in that they enable the high-mass loading of Ni sites. However, thermal stripping and exfoliation to derive porous structures appear to be on the very well-known conventional processes.

Reply: Thank you very much for your constructive comments. We also thanks for your recognition of the synthetic strategy for high-loading single-atom catalysts (g- C_3N_4 -baesd materials) with porous ultrathin structures. We respond to the reviewer's high-quality comments **by explaining the synthetic strategy** for the high-loading catalyst in more detail and **supplementing the relevant experiments**.

1. Interpretation and illustration of the synthetic strategy for high-loading M-SAPs and corresponding revisions in the manuscript.

1.1 Regarding the derivation of porous ultrathin g- C_3N_4 from bulky g- C_3N_4 structures (BCN), we acknowledge that the process of thermal stripping and exfoliation to achieve porous structures is indeed a well-known conventional method. Despite the conventional nature of the

process, we found the results to be highly promising in terms of enabling the high-loading of Ni single atom sites, which significantly enhances the photocatalytic activity for H₂O₂ production. In terms of synthetic methods, we hope to provide a simple and efficient high-loading single-atom synthesis strategy for the reference of researchers.

In fact, the highly loaded single-atom catalysts proposed in our manuscript are a **two-step synthesis strategy**, rather than relying solely on porous ultrathin structures. ① The first step is to adjust the microscopic morphology of the catalyst (manufacturing a **porous ultrathin structure**); ② **the second step is** to optimize the single-atom loading process (**critical continuous ultrasonic treatment** in wet-chemical precipitation). In the wet chemical precipitation method, the continuous ultrasonic treatment proposed in our manuscript is not only beneficial to further thinning the g-C₃N₄ to provide more loading sites, but also conducive to the highly dispersed and anchored metal ions on the g-C₃N₄ substrate.

However, this may be caused by the unclear expression of the synthesis strategy in the previous manuscript, and we have revised the relevant description as follows (the modified content is highlighted in yellow):

A schematic diagram of the synthesis of high-loading M-SAPs is shown in **Fig. 1a**, along with the presumed structural changes of the heptazine unit in the corresponding g-C₃N₄. Briefly, the synthesis of high-loading M-SAPs is mainly divided into 2 steps: regulating the microtopography and further optimizing the loading process (**continuous ultrasonic treatment in wet-chemical precipitation**). Firstly, the substrate microtopography was adjusted by thermal stripping and ultrasonic exfoliation of the original bulk g-C₃N₄ (denoted as **BCN**), thereby preparing porous ultrathin g-C₃N₄ nanosheets (denoted as **PuCN**), which is beneficial to providing more single atoms loading sites. Next, PuCN was further loaded with metal single atoms by wet-chemical precipitation under continuous ultrasonic conditions. The continuous ultrasonic treatment can not only promote the uniform dispersion of single atoms, but also further exfoliate g-C₃N₄ (destroy the van der Waals forces between carbon nitride layers) to provide abundant loading sites, and finally achieve a high-loading M-SAPs with porous ultrathin structure (denoted as **M_{SAPs}-PuCN**, see **Methods** for details).

1.2 The porous material has a large specific surface area and rich pore distribution, which increases the contact area with the reactants and facilitates the shuttle of the reactants in the pores to promote the improvement of activity. In terms of material design, high-loading single-atom catalysts with thin-layer porous structures are extremely advantageous for the realization of high catalytic activity in our manuscript. Meanwhile, the development of single-atom catalysts with porous structures has received further recognition and attention in the field of photo/electrocatalysis recently (*Nature Synthesis*, 10.1038/s44160-022-00129-x, **2022**; *Chem*, 10.1016/j.chempr.2023.06.021, **2023**).

2. Supplemented relevant experimental.

① Guided by the reviewer's constructive comments, in order to further illustrate that continuous ultrasonic treatment is beneficial to increase the loading of single atoms, **we**

supplemented the spherical aberration electron microscope characterization of Ni_{SAPs}-PuCN under different ultrasonic times to visually observe the loading and distribution of the single atoms.

We followed the synthesis method in the manuscript, only changing the time of sonication (the content of NiCl₂·6H₂O added was equal), and synthesized Ni_{SAPs}-PuCN samples under different sonication times. The ultrasonic treatment time was 1 h, 2 h, and 3 h, and the corresponding samples were named as Ni_{SAPs}-PuCN-1h, Ni_{SAPs}-PuCN-2h, and Ni_{SAPs}-PuCN-3h. We supplemented the above experimental characterization and analysis in **Supplementary Information (Supplementary Figure 37)**, as shown in **Figure 12** below (the modified content is highlighted in yellow):

Figure 12 (Supplementary Figure 37). Aberration-corrected HAADF-STEM image of (a) Ni_{SAPs}-PuCN-1h, (b) Ni_{SAPs}-PuCN-2h, and (c) Ni_{SAPs}-PuCN-3h (the yellow dotted lines represent holes in the sample).

The Ni single-atom samples synthesized under different continuous ultrasonic treatment times (1 h, 2 h, and 3 h) were named Ni_{SAPs}-PuCN-1h, Ni_{SAPs}-PuCN-2h, and Ni_{SAPs}-PuCN-3h in turn. As shown in **Supplementary Figure 37**, with the prolongation of the continuous ultrasonic time (1 h, 2 h, and 3 h), it can be seen intuitively under the spherical aberration transmission electron microscope that Ni single atoms are more uniformly dispersed on the porous structure and the loading content of Ni single atoms increases significantly. Meanwhile, the inductively coupled plasma mass spectrometry (ICP-MS) further verified that the Ni single atom loadings in Ni_{SAPs}-PuCN-1h, Ni_{SAPs}-PuCN-2h, and Ni_{SAPs}-PuCN-3h were 3.7 wt%, 8.3 wt%, and 12.5 wt%, respectively. Therefore, continuous ultrasonic treatment can effectively promote the uniform distribution of high-loading Ni single atoms on porous ultrathin g-C₃N₄.

② The above has been added to the Method section of the revised manuscript, with corresponding changes as follows (the modified content is highlighted in yellow):

Then, under continuous ultrasonic and stirring, a certain amount of NiCl₂·6H₂O solution (1.5 mg ml⁻¹; water to ethanol is 1:1) was slowly added dropwise, and ultrasonically treated for 3 h to ensure full contact between metal ions and PuCN (continuous sonication is beneficial to promote uniform dispersion and high loading of single atoms, see **Supplementary Fig. 37** for more details).

Reviewer #3-Comment 2:

Besides, the demonstration of single metal atoms should be elaborate.

Reply: We appreciate your keen observation and insightful comments regarding the demonstration of single metal atoms in our research. The theme of our manuscript is to reveal the relationship between the in situ dynamic structure evolution of Ni single-atom sites on g-C₃N₄ and the high activity and high selectivity in photocatalytic H₂O₂ production reactions. The demonstration of Ni single atoms on g-C₃N₄ is particularly important. In previous manuscripts, the demonstration of Ni single atoms has been jointly demonstrated by aberration-corrected HAADF-STEM (Fig. 2 of the manuscript) and XAFS (Fig. 3 of the manuscript) characterization. According to the constructive comment of the reviewer, in order to prove Ni single atoms more comprehensively, we supplemented the more detailed analysis of the aberration-corrected HAADF-STEM image of Ni_{SAPs}-PuCN. The above content and analysis are organized in **Supplementary Information (Supplementary Figure 14)**, as shown in **Figure 13** below (the modified content is highlighted in yellow):

Figure 13 (Supplementary Figure 14) (a) Aberration-corrected HAADF-STEM image of Ni_{SAPs}-PuCN. (b-c) The intensity profile obtained from the Ni single atom sites (Site 1 and Site 2).

As shown in **Supplementary Figure 14**, we selected typical two site regions (site 1 and site 2) in the aberration-corrected HAADF-STEM image of Ni_{SAPs}-PuCN for intensity analysis. The contrast strength of atoms in aberration-corrected HAADF-STEM images strongly depends on atomic number (*Nat. Nanotechnol.* 2022, 17, 590-602). In **Supplementary Figure 14b-c**, it can be very intuitively seen that the intensity of single bright spot in the two-site region is significantly higher than that of the base structure (CN structure), which confirms that the Ni single-atom sites are dispersed in the g-C₃N₄.

Meanwhile, in the revised manuscript, the corresponding changes are as follows (the modified

content is highlighted in yellow):

This adjusted structure would be very favorable for anchoring single atoms, which was confirmed in the subsequent HAADF-STEM characterization (Fig. 2e and 2g). In Fig. 2e, numerous isolated bright spots clearly differ from the contrast of the g-C₃N₄ substrate (See **Supplementary Fig. 14** for more details), indicating that the high density of Ni single atoms was successfully dispersed on the Ni_{SAPs}-PuCN without any nanoparticles or clusters being found. The EDS mapping (Fig. 2f) showed that Ni, C and N elements were uniformly distributed on the ultrathin porous substrate.

Reviewer #3-Comment 3:

Moreover, the authors need to demonstrate that single atoms are maintained after the repeated photocatalytic cycles as well as long-hour operation.

Reply: Thanks for your comments and valuable suggestions. We supplemented Ni_{SAPs}-PuCN sample after long reaction cycles (8 reaction cycles, 8 h) using X-ray absorption fine structure spectroscopy (XAFS), which proved that Ni_{SAPs}-PuCN still maintains the Ni single-atom state after long-hour reaction cycles. To further ensure the accuracy, the Ni_{SAPs}-PuCN sample after testing XAFS were re-characterized by aberration-corrected transmission electron microscopy, which also confirmed that Ni single atoms in Ni_{SAPs}-PuCN are stable after 8 cycles of reaction. In the previous manuscript (Supplementary Figure 22 of the previous manuscript; Supplementary Figure 24 of the revised manuscript), the stability of Ni_{SAPs}-PuCN photocatalytic H₂O₂ performance in 8 cycles reaction has been demonstrated. The XAFS data and electron microscopy characterization data of the Ni_{SAPs}-PuCN after 8 cycles reaction were added in **Supplementary Figure 25c-f** to further verify the structural stability of Ni_{SAPs}-PuCN, and the modified **Supplementary Figure 25** and analysis are shown in **Figure 14** below (the modified content is highlighted in yellow):

Figure 14 (Supplementary Figure 25). Structural stability of Ni_{SAPs}-PuCN after photocatalytic 8-cycle reaction. The (a) XRD and (b) FTIR spectrum of Ni_{SAPs}-PuCN after the photocatalytic reaction. (c) Ni K-edge XANES spectra of the Ni_{SAPs}-PuCN sample before and after the 8 cycles of reaction. (d) Fourier transform of EXAFS spectra of Ni_{SAPs}-PuCN before and after 8 cycles of reaction. (e) EDS mapping and (f) HAADF-STEM image of Ni_{SAPs}-PuCN after 8 cycles of reaction.

The XRD and FTIR spectra of Ni_{SAPs}-PuCN (Supplementary Figure 25a-b) did not change much before and after the photocatalytic reaction, and the 8-cycle performance did not weaken significantly (Supplementary Figure 24), indicating that Ni_{SAPs}-PuCN has good catalytic cycle and stability. Further, the stability of Ni single atoms in Ni_{SAPs}-PuCN after the 8 cycles reaction

was confirmed by X-ray absorption fine structure spectroscopy (XAFS). As shown in **Supplementary Fig. 25c**, the Ni K-edge X-ray absorption near-edge structure (XANES) spectra of Ni_{SAPs}-PuCN before and after the reaction were basically changed little, proving that the valence state of Ni in Ni_{SAPs}-PuCN was basically unchanged before and after the reaction. Meanwhile, **Supplementary Fig. 25d** shows the Fourier transform of the extended X-ray absorption fine structure (FT-EXAFS) spectra of Ni_{SAPs}-PuCN before and after the reaction. It can be seen in **Supplementary Fig. 25d** that the Ni_{SAPs}-PuCN after 8 cycles reaction also exhibits a main peak around 1.69 Å (Ni-N), and there is no Ni-Ni bond in Ni_{SAPs}-PuCN after reaction. The EDS mapping (**Supplementary Figure 25e**) showed that the Ni element was still uniformly dispersed in Ni_{SAPs}-PuCN after 8 cycles of reaction. Moreover, the Ni single atoms were still highly dispersed on Ni_{SAPs}-PuCN after the cycling reaction (**Supplementary Figure 25f**), and no Ni nanoclusters or particles were found. In summary, the above characterizations fully proved that Ni single atoms in Ni_{SAPs}-PuCN after long-hour cyclic reaction are stable, indicating the good catalytic cycle and stability of Ni_{SAPs}-PuCN.

Reviewer #3-Comment 4:

Also, the quantum yield at 420 nm that is not the major spectrum of sun light, is considered to be not making a great breakthrough.

Reply: Thanks for your constructive comments. **First of all, we respect your comment and thank you for your kind reminder.** Second, **we carefully answer the reviewer's question by explaining g-C₃N₄ materials, current evaluation metrics for photocatalytic H₂O₂ performance, and the excellent H₂O₂ production activity of Ni_{SAPs}-PuCN.** Respond carefully from the following four points:

① The g-C₃N₄ is defined as a visible light catalytic material since it was developed¹. The visible light spectrum wavelength ranges from about 400 ~ 420 nm to 750 nm, and the band structure of g-C₃N₄ enables it to absorb photons in this range and generate photogenerated electron-hole pairs¹. For visible light photocatalysts, the ability to efficiently absorb light and carry out catalytic reactions in this wavelength range is the key to realizing visible light-driven photocatalytic reactions¹⁻³. As the reviewer mentioned, **420 nm is not the main wavelength range of the solar spectrum, but it is still included in the spectral range of visible light.** In numerous photocatalytic research works as well as practical applications, such as photocatalytic H₂O₂ production, water splitting or CO₂ reduction, etc., it is crucial to be able to efficiently utilize the wavelength range of visible light from the sun light⁴⁻⁸. Therefore, evaluating the performance of photocatalysts against the AQY at 420 nm (or 400 nm) can provide information about their activity in the visible light range, which is also an important evaluation index for numerous high-quality photocatalytic research efforts^{3, 5, 7}. We illustrate with the following example:

- (a) Teng et al. developed Sb-SAPC catalysts for the photocatalytic H₂O₂ production under visible light. In the “Abstract” of this paper (*Nature Catalysis*, 10.1038/s41929-021-00605-1, 2021), the AQY at 420 nm was used to describe the photocatalytic activity: An apparent quantum yield of 17.6% at 420 nm together with a solar-to-chemical conversion efficiency of 0.61% for H₂O₂ synthesis was achieved.

(b) Zhao et al. developed CNN/BDCNN (*Nature Energy*, 10.1038/s41560-021-00795-9, 2021) catalysts for photocatalytic water splitting under visible light. In this paper, the **AQY at 400 nm** was used to describe the photocatalytic activity:

The AQY at 400 nm is calculated to be 5.95% by considering the whole reaction as a one-step excitation process, or 11.90% as defined for the Z-scheme system, which surpasses the AQY values of most previously reported g-C₃N₄-based photocatalysts.

② In our manuscript, the AQY at 420 nm of Ni_{SAPs}-PuCN for photocatalytic H₂O₂ production in pure water reached 10.9%, indeed exceeding the vast majority of g-C₃N₄-based photocatalysts reported so far. At the same time, compared with other types catalytic materials currently developed for photocatalytic H₂O₂ production, the AQY value of Ni_{SAPs}-PuCN is not inferior. To further illustrate, we compared the AQY of Ni_{SAPs}-PuCN with those of previous paper (Supplementary Figure 21 of the previous manuscript, Supplementary Figure 23 of the revised manuscript), as shown in **Figure 15** below:

Figure 15 (Supplementary Figure 23 of the revised manuscript). Summarized AQY at 420 nm of recently reported photocatalysts (g-C₃N₄-based and other types of photocatalysts) for H₂O₂ production in pure water. The comparison details of all the above samples are collated in Supplementary Table 4.

③ In order to comprehensively measure the photocatalytic H₂O₂ activity of Ni_{SAPs}-PuCN, in addition to the AQY index, the standard H₂O₂ yield ($\mu\text{mol g}^{-1} \text{h}^{-1}$) and the solar-to-chemical conversion (SCC) efficiency are also required. In our manuscript, the H₂O₂ production rate of Ni_{SAPs}-PuCN in pure water under visible light is 342.2 $\mu\text{mol g}^{-1} \text{h}^{-1}$, the AQY at 420 nm is 10.9%, and the SCC efficiency is 0.82%. With these comprehensive photocatalytic performance test indicators, Ni_{SAPs}-PuCN has excellent H₂O₂ production activity, which is the most efficient g-C₃N₄-based photocatalyst for H₂O₂ production reported so far. In Supplementary Table 4 of the previous manuscript, we sorted out 25 related research works, and further illustrated our advantages in performance through systematic comparison, as shown in Figure 16 below (due to the introduction of new references in the revised Supplementary Information, the order of the references corresponding to each sample in the revised Supplementary Table 4 has been

modified):

catalyst [Ⓢ]	Reaction solution and catalytic concentration [Ⓢ]	Light Source [Ⓢ]	H ₂ O ₂ [Ⓢ] (μmol g ⁻¹ h ⁻¹) [Ⓢ]	AQY [Ⓢ] at 420 nm [Ⓢ]	SCC [Ⓢ]	Ref. [Ⓢ]
NiSAP₈-PuCN[Ⓢ]	Pure water (1 g L⁻¹)[Ⓢ]	λ ≥ 420 nm[Ⓢ]	342.2[Ⓢ]	10.9%[Ⓢ]	1.17% (1h)[Ⓢ] 0.90% (2h)[Ⓢ] 0.82% (3h)[Ⓢ]	This work[Ⓢ]
g-C ₃ N ₄ /PDI [Ⓢ]	Pure water (1.66 g L ⁻¹) [Ⓢ]	λ ≥ 420 nm [Ⓢ]	21 [Ⓢ]	2.5% [Ⓢ]	0.1% (2h) [Ⓢ]	15 [Ⓢ]
g-C ₃ N ₄ /BDI [Ⓢ]	Pure water (1.66 g L ⁻¹) [Ⓢ]	λ ≥ 420 nm [Ⓢ]	34 [Ⓢ]	4.6% [Ⓢ]	0.13% (2h) [Ⓢ]	16 [Ⓢ]
g-C ₃ N ₄ /MTI [Ⓢ]	Pure water (1.66 g L ⁻¹) [Ⓢ]	λ ≥ 420 nm [Ⓢ]	22 [Ⓢ]	6.1% [Ⓢ]	0.18% (2h) [Ⓢ]	17 [Ⓢ]
g-C ₃ N ₄ /PDI/rGO [Ⓢ]	Pure water (1.66 g L ⁻¹) [Ⓢ]	λ ≥ 420 nm [Ⓢ]	24.1 [Ⓢ]	6.1% [Ⓢ]	0.2% (2h) [Ⓢ]	18 [Ⓢ]
g-C ₃ N ₄ /PDI-BN-rGO [Ⓢ]	Pure water (1.66 g L ⁻¹) [Ⓢ]	λ ≥ 420 nm [Ⓢ]	29.1 [Ⓢ]	7.3% [Ⓢ]	0.28% (2h) [Ⓢ]	19 [Ⓢ]
Nv-C≡N-CN [Ⓢ]	Pure water (1 g L ⁻¹) [Ⓢ]	λ ≥ 420 nm [Ⓢ]	137 [Ⓢ]	1.8% [Ⓢ]	0.23% (1h) [Ⓢ]	7 [Ⓢ]
ZnPPC-NBCN [Ⓢ]	Pure water (0.4 g L ⁻¹) [Ⓢ]	λ ≥ 420 nm [Ⓢ]	114 [Ⓢ]	N.T. [Ⓢ]	N.T. [Ⓢ]	20 [Ⓢ]
CPN [Ⓢ]	Pure water (0.66) [Ⓢ]	λ ≥ 420 nm [Ⓢ]	246 [Ⓢ]	1.57% [Ⓢ]	0.43% (8h) [Ⓢ]	21 [Ⓢ]
Ag@U-g-C ₃ N ₄ [Ⓢ]	Pure water (1 g L ⁻¹) [Ⓢ]	λ ≥ 420 nm [Ⓢ]	70 [Ⓢ]	N.T. [Ⓢ]	N.T. [Ⓢ]	22 [Ⓢ]
Cv-g-C ₃ N ₄ [Ⓢ]	Pure water (1 g L ⁻¹) [Ⓢ]	λ ≥ 420 nm [Ⓢ]	90 [Ⓢ]	N.T. [Ⓢ]	N.T. [Ⓢ]	23 [Ⓢ]
3DM-g-C ₃ N ₄ -PW [Ⓢ]	Pure water (1 g L ⁻¹) [Ⓢ]	λ ≥ 350 nm [Ⓢ]	35 [Ⓢ]	N.T. [Ⓢ]	N.T. [Ⓢ]	24 [Ⓢ]
HJ-C ₃ N ₄ [Ⓢ]	Pure water (1 g L ⁻¹) [Ⓢ]	λ ≥ 420 nm [Ⓢ]	115 [Ⓢ]	N.T. [Ⓢ]	N.T. [Ⓢ]	25 [Ⓢ]
K-g-C ₃ N ₄ -NH-CH ₂ -OH [Ⓢ]	Pure water (1 g L ⁻¹) [Ⓢ]	λ ≥ 420 nm [Ⓢ]	30.4 [Ⓢ]	N.T. [Ⓢ]	0.29% (1h) [Ⓢ]	26 [Ⓢ]
Co/AQ/C ₃ N ₄ [Ⓢ]	Pure water (0.5 g L ⁻¹) [Ⓢ]	AM 1.5G [Ⓢ]	124 [Ⓢ]	0.054% [Ⓢ]	0.014%(1h) [Ⓢ]	27 [Ⓢ]
G-CN-PWO [Ⓢ]	Pure water (1 g L ⁻¹) [Ⓢ]	λ ≥ 420 nm [Ⓢ]	60 [Ⓢ]	N.T. [Ⓢ]	N.T. [Ⓢ]	28 [Ⓢ]
5Cv@g-C ₃ N ₄ [Ⓢ]	Pure water (1 g L ⁻¹) [Ⓢ]	λ ≥ 420 nm [Ⓢ]	124.5 [Ⓢ]	N.T. [Ⓢ]	N.T. [Ⓢ]	29 [Ⓢ]
R370-CN [Ⓢ]	Pure water (1 g L ⁻¹) [Ⓢ]	λ ≥ 420 nm [Ⓢ]	170 [Ⓢ]	4.3% [Ⓢ]	0.26% (1h) [Ⓢ]	30 [Ⓢ]
Sb-SACS [Ⓢ]	Pure water (2 g L ⁻¹) [Ⓢ]	λ ≥ 420 nm [Ⓢ]	196 [Ⓢ]	17.6% [Ⓢ]	0.61% (4h) [Ⓢ]	14 [Ⓢ]
RF523 [Ⓢ]	Pure water (1.66 g L ⁻¹) [Ⓢ]	λ ≥ 420 nm [Ⓢ]	52 [Ⓢ]	8% [Ⓢ]	0.5% (5h) [Ⓢ]	31 [Ⓢ]
C ₂ N ₅ [Ⓢ]	Pure water (10 g L ⁻¹) [Ⓢ]	λ ≥ 420 nm [Ⓢ]	155 [Ⓢ]	15.4% [Ⓢ]	0.55% (6h) [Ⓢ]	32 [Ⓢ]
CTF-BDDBN [Ⓢ]	Pure water (0.6 g L ⁻¹) [Ⓢ]	λ ≥ 420 nm [Ⓢ]	97 [Ⓢ]	N.T. [Ⓢ]	0.14% (1h) [Ⓢ]	33 [Ⓢ]
DE7-M [Ⓢ]	Pure water (1.66 g L ⁻¹) [Ⓢ]	λ ≥ 420 nm [Ⓢ]	221.6 [Ⓢ]	8.7% [Ⓢ]	0.23% (5h) [Ⓢ]	34 [Ⓢ]
CHF-DPDA [Ⓢ]	Pure water (2 g L ⁻¹) [Ⓢ]	λ ≥ 420 nm [Ⓢ]	256 [Ⓢ]	16% [Ⓢ]	0.78% (1h) [Ⓢ]	35 [Ⓢ]
RF-DHAQ-2 [Ⓢ]	Pure water (0.2 g L ⁻¹) [Ⓢ]	λ ≥ 420 nm [Ⓢ]	1820 [Ⓢ]	11.6% [Ⓢ]	1.2% (1h) [Ⓢ]	36 [Ⓢ]
RF/P3HT-1.0 [Ⓢ]	Pure water (1.66 g L ⁻¹) [Ⓢ]	λ ≥ 420 nm [Ⓢ]	123 [Ⓢ]	10.5% [Ⓢ]	1.0% (3h) [Ⓢ]	37 [Ⓢ]

Figure 16 (Supplementary Table 4). Performance comparison of recently reported materials for photocatalytic production of H₂O₂ in pure water. The materials of the blue background are g-C₃N₄-based photocatalyst, and the materials of the yellow background are other types of photocatalysts. The standardized yield of H₂O₂ (μmol g⁻¹ h⁻¹), AQY, and SCC efficiency are used as evaluation indicators.

In addition, we have revised the inappropriate wording for performance description in the manuscript, and changed "leading" to "excellent", as follows:

Overall, according to the standard H₂O₂ yield, AQY and SCC efficiency as the evaluation indicators of photocatalytic activity, NiSAP₈-PuCN has excellent leading H₂O₂ generation activity in pure water.

④ In **Figure 4b** in our manuscript, we noted and compared the effects of visible light ($\lambda \geq 420$ nm) and simulated sunlight (AM 1.5G) on the photocatalytic activity of Ni_{SAPs}-PuCN. Under visible light ($\lambda \geq 420$ nm), the photocatalytic H₂O₂ yield of Ni_{SAPs}-PuCN was 346.2 $\mu\text{mol g}^{-1} \text{h}^{-1}$; under simulated sunlight, the photocatalytic yield of Ni_{SAPs}-PuCN increased to 640.1 $\mu\text{mol g}^{-1} \text{h}^{-1}$. Under simulated sunlight irradiation, g-C₃N₄ materials can absorb shorter-wavelength ultraviolet rays and longer-wavelength infrared rays than under visible light irradiation. This enables Ni_{SAPs}-PuCN to obtain more light energy under simulated sunlight, so its photocatalytic activity is significantly improved. Under the strict pursuit of visible light-driven photocatalysis in the current g-C₃N₄-based material system, the higher activity of Ni_{SAPs}-PuCN under simulated sunlight also reflects its performance advantages.

Thanks again for the reviewer's constructive comment, and we hope that the above sincere and conscientious responses fully illustrate the performance advantages of Ni_{SAPs}-PuCN in the manuscript.

Reviewer #3-Comment 5:

The solar-to-chemical conversion conversion (0.82%) is not much superior to the previous results (Figure 4g).

Reply: Thanks for your comment and kind reminder. The reviewer is right. In **Figure 4g**, **although** the solar-to-chemical conversion (SCC) efficiency (0.82%) of Ni_{SAPs}-PuCN in pure water for photocatalytic H₂O₂ synthesis is the highest reported for g-C₃N₄-based materials so far, **our SCC efficiency indeed does not increase exponentially compared with previous research work.**

The SCC efficiency of the photocatalyst directly reflects its efficiency in converting solar energy into chemical energy, and the development of highly active photocatalysts to achieve high SCC efficiency will benefit subsequent practical applications. However, this is also one of the major challenges facing the entire field of photocatalysis. **The improvement of SCC efficiency currently reported in photocatalysis is also relatively slow, and the same is true for the photocatalytic H₂O₂ production.** We further illustrate by enumerating the SCC efficiencies of g-C₃N₄-based photocatalytic H₂O₂ production published in recent years:

- (a) In **2021**, Teng et al. developed the Sb-SAPC catalyst with a SCC efficiency of 0.61% for the synthesis of H₂O₂ in pure water, which was the **highest SCC efficiency reported for g-C₃N₄-based materials that year** (*Nature Catalysis*, 10.1038/s41929-021-00605-1, 2021).
- (b) In **2023**, Tian et al. developed the CNIO-GaSA catalyst with excellent H₂O₂ production activity and achieved SCC efficiency of 0.4% (*Nature Synthesis*, 10.1038/s44160-023-00272-z, 2023).

Thanks to the reviewer's nice comment, we hope to develop highly efficient photocatalysts to achieve a greater increase in SCC efficiency in our follow-up research work.

Reviewer #3-Comment 6:

Besides, the quantum yields in the visible-light spectra over 450 nm are shown to be significantly degrading, as shown in Figure 4c.

Reply: We appreciate your careful evaluation of the AQY results and this high-quality comment. The reviewer is right. We acknowledge that the AQY efficiency of NiSAPs-PuCN gradually decreases with increasing wavelength (Figure 4c of the manuscript). However, this is a critical issue commonly faced in current photocatalytic systems, and this trend appears not only in our results but also in the vast majority of photocatalytic research articles^{1, 4, 5, 7}. We compared Figure 4c in the manuscript with photocatalytic AQY data published in a series of high-quality journals to illustrate this trend, as shown in Figure 17 below.

Figure 17. (a) The wavelength-dependent AQY for photocatalytic H₂O₂ production in pure water by NiSAPs-PuCN (Figure 4c of the manuscript). (b) The AQY of CNN/BDCNN (Z-type heterojunction g-C₃N₄; *Nature Energy*, 10.1038/s41560-021-00795-9, 2021) in photocatalytic overall water splitting. (c) The AQY of Sb-SAPC-15 (Sb single atoms in g-C₃N₄, *Nature Catalysis*, 10.1038/s41929-021-00605-1, 2021) in photocatalytic H₂O₂ production. (d) The AQY of SA-TCPP supramolecular (*Nature Energy*, 10.1038/s41560-023-01218-7, 2023) in photocatalytic H₂O₂ production. (e) The AQY of CNIO-GaSA (Ga single atoms in g-C₃N₄, *Nature Synthesis*, 10.1038/s44160-023-00272-z, 2023) in photocatalytic H₂O₂ production.

As shown in Figure 17b-e, compared with our AQY data trend (Figure 17a), regardless of the type of photocatalyst (g-C₃N₄-based materials or supramolecular materials), and no matter what kind of photocatalytic reaction (photocatalytic H₂O₂ production or overall water splitting), the apparent quantum efficiency (AQY) of most photocatalysts would decrease significantly with the increase of wavelength, especially after 450 nm. Inspired by the valuable comments of the reviewer, we carefully analyzed the approximate reasons for this ubiquitous phenomenon, as follows:

① **Light absorptivity:** The light absorptivity of photocatalysts generally decreases with increasing wavelength (as shown in **Figure 14**). In the visible range, photocatalysts tend to exhibit excellent light absorption at shorter wavelengths but weaker absorption at longer wavelengths. Consequently, this would lead to a decrease in photocatalytic activity at longer wavelengths, leading to a gradual decrease in AQY of the photocatalyst.

② **Band structure:** The band structure of the photocatalyst determines its electronic behavior under photoexcitation. As the wavelength increases, the energy of the incident light will gradually decrease, which may fail to fully excite the electrons inside the photocatalyst, thereby reducing the AQY of the photocatalyst.

Thanks again to the reviewer's high value comment. As the wavelength increases, the AQY of most photocatalysts gradually decreases, which indeed restricts the subsequent practical application of photocatalytic technology. Inspired by this reviewer, we further deepened our understanding of this issue, and in our follow-up work we hope to develop strategies to overcome these challenges and improve the performance of photocatalytic.

Reviewer #3-Comment 7:

Besides, the simultaneous mechanisms for exciton (electron-hole pair) transfer involving in the photocatalytic reduction of O_2 into H_2O_2 are unclear.

Reply: Thanks for your nice comment and valuable information. In our previous manuscript, the mechanism of photogenerated electron-hole transfer in the photocatalytic H_2O_2 reaction was indeed deficient. We added a series of experimental characterizations (① **transient photocurrent response** and ② **femtosecond transient absorption spectroscopy**) to supplemented the separation/transportation and kinetic behaviors mechanism of photogenerated carriers (electron-hole pairs), and further ③ combined with the surface reaction mechanism (O_2 to H_2O_2) to illustrate this key content.

① We **supplemented the transient photocurrent response (TPR) measurements** of the samples and organized the results in the revised Supplementary Information (**Supplementary Figure 27d**). The supplementary TPR, combined with the previous PL, TPPL and EIS spectra (**Supplementary Figure 27** of the revised **Supplementary Information**), systematically reveals that the introduction of Ni single atoms greatly promotes the separation and transport characteristics of photogenerated carriers, which is conducive to promoting the occurrence of subsequent surface reactions (O_2 to H_2O_2). The modified **Supplementary Figure 27** and analysis are shown in **Figure 18** below (the modified content is highlighted in yellow):

Figure 18 (Supplementary Figure 27). (a) Photoluminescence (PL) spectra, (b) electrochemical chemical impedance spectra (EIS), (c) time-resolved photoluminescence (TRPL) spectra, and (d) transient photocurrent response (TPR) of BCN, PuCN, and Ni_{SAPs}-PuCN.

In Supplementary Figures 27a-b, the Ni_{SAPs}-PuCN exhibited the lowest luminescence intensity and the smallest interfacial resistance, which indicated Ni single atoms suppressed the recombination of carriers and facilitated the carrier transport. Meanwhile, TRPL (Supplementary Fig. 27c) showed the same trend, in which the average radiation lifetimes of BCN, PuCN and Ni_{SAPs}-PuCN were 5.29, 4.61, and 2.52 ns (fitting date in Supplementary Table 7) respectively, implying that the Ni-N₃ sites can rapidly trap electrons and accelerate electron transport. In Supplementary Figure 27d, compared to BCN and PuCN, Ni_{SAPs}-PuCN exhibited a stronger photocurrent density after turning on the light, indicating that the introduction of Ni single atoms facilitated the transport of photogenerated carriers.

② In order to further reveal the kinetic behaviors of photogenerated carriers induced by Ni single atoms, more refined femtosecond transient absorption spectroscopy (fs-TAS) measurements were performed on BCN and Ni_{SAPs}-PuCN, and the results and analysis are added to the revised Supplementary Information (Supplementary Figure 28), as shown in Figure 19 below, as follows:

Figure 18 (Supplementary Figure 28). Visible transient absorption spectra measurements of (a) BCN and (b) NiSAPs-PuCN with a 400 nm laser flash. (c) Decay kinetics of photogenerated carriers fitted through a double-exponential function (The inset table are fitted lifetimes).

To further monitor the kinetic behaviors of photogenerated carriers, femtosecond transient absorption spectroscopy (fs-TAS) were performed on BCN and NiSAPs-PuCN upon 400 nm laser excitation. The fs-TA data of BCN and NiSAPs-PuCN in water are shown in **Supplementary Figure 28a-b**. The BCN (**Supplementary Figure 28a**) and NiSAPs-PuCN (**Supplementary Figure 28b**) have broad negative signals from 420 to 650 nm, which are attributed to ground state bleaching and stimulated emission^{9, 10}. The decay kinetics of the photo-excited carriers of BCN and NiSAPs-PuCN were probing at 540 nm and fitting through a double-exponential function, as shown in **Supplementary Figure 28c**. The τ_1 and τ_2 in **Supplementary Figure 28c** are designated as the shallow trapping of electrons and the recombination of shallowly trapped electrons, respectively. The average lifetimes (τ_{avr}) of BCN and NiSAPs-PuCN after fitting are 2.56 and 1.27 ps, respectively. The slower time scale of intrinsic deep trapped states in BCN corresponds to a longer lifetime, which has been shown to lead to poorer photocatalytic activity^{7, 10}. Obviously, due to the introduction of Ni single atoms, NiSAPs-PuCN has a shorter lifetime, which could be attributed to the generated deeply trapped sites and has been demonstrated to contribute to the $2e^-$ oxygen reduction reaction (ORR) process^{7, 11}, thus facilitating the accelerated reduction of O_2 to H_2O_2 .

Based on this reviewer's constructive comments, we have added the above results and analysis to the revised manuscript as follows (the modified content is highlighted in yellow):

The introduction of Ni single atoms effectively promotes carrier separation and transport, as confirmed by a series of spectroscopic measurements (**Supplementary Fig. 27**). Next, femtosecond transient absorption spectroscopy (fs-TAS) was used to further reveal the kinetic behaviors of the photogenerated carriers. The results indicate that Ni_{SAPs}-PuCN has a shorter lifetime than BCN (See **Supplementary Fig. 28** for more details), which could be attributed to the deep trapping sites induced by Ni single atoms and has been demonstrated to facilitate the 2e⁻ ORR process.

③ The photocatalytic mechanism is generally divided into three stages: light absorption, photogenerated carrier separation and transport, and surface reaction^{1,2,3,4}.

3.1 In the photogenerated carrier separation and transport stage, through the above **Supplementary Figure 27** and **Supplementary Figure 28**, it is verified experimentally that the introduction of Ni single atoms greatly optimizes the carrier separation and transport characteristics. Moreover, the electronic structure of the photocatalyst greatly affects its photogenerated carrier separation/transport properties and subsequent surface reactions^{3,5,7}. Combined with the theoretical calculations in the manuscript (**Supplementary Figures 27-29** in the previous manuscript; **Supplementary Figures 30-32** in the revised manuscript), the Ni 3d electrons in Ni_{SAPs}-PuCN greatly optimize the electronic structure of g-C₃N₄, which is closely related to the enhanced carrier separation and transport properties.

3.2 In the surface reaction stage, the photocatalytic H₂O₂ production activity (**Figure 4**), in situ reaction mechanism (**Figure 5**), and theoretical calculation (**Figure 6**) sections of the manuscript illustrated that Ni single-atom sites in Ni_{SAPs}-PuCN are the reaction centers of 2e⁻ ORR.

Therefore, combined with the supplementary experiments (**Supplementary Figures 27-28**) above and the entire manuscript, we summarize the photocatalytic reaction mechanism of Ni_{SAPs}-PuCN to carefully answer the reviewer's questions:

First, the Ni_{SAPs}-PuCN generated photogenerated carriers (electron-hole pairs) under visible light irradiation. Next, benefiting from the good carrier separation and transport properties, the separated photogenerated electrons are mainly concentrated on Ni single atomic sites for O₂ reduction reaction (O₂ → H₂O₂). The separated photogenerated holes have been confirmed to oxidize water to generate O₂ (H₂O → O₂), which is consistent with the current reported photocatalytic 2e⁻ ORR literature^{7,11}. Focusing on the reduction reaction of O₂ to H₂O₂, we revealed that the dynamic structure evolution (Ni-N₃ to O₁-Ni-N₂ to HOO-Ni-N₂) of Ni single atomic sites is closely related to its high H₂O₂ activity (O₂ → ·O₂ → ·OOH → H₂O₂) through a series of spectroscopic characterizations (such as in situ XAFS) and theoretical calculations.

Thanks again to this reviewer for the constructive comments and valuable suggestions to guide our work and further improve the quality of the manuscript.

References

1. Wang X, *et al.* A metal-free polymeric photocatalyst for hydrogen production from water under visible light. *Nat Mater* **8**, 76-80 (2009).
2. Chen R, *et al.* Charge separation via asymmetric illumination in photocatalytic Cu₂O particles. **3**, 655-663 (2018).
3. Wang Y, *et al.* Current understanding and challenges of solar-driven hydrogen generation using polymeric photocatalysts. *Nature Energy* **4**, 746-760 (2019).
4. Zhang Y, *et al.* H₂O₂ generation from O₂ and H₂O on a near-infrared absorbing porphyrin supramolecular photocatalyst. *Nature Energy*, (2023).
5. Zhao D, *et al.* Boron-doped nitrogen-deficient carbon nitride-based Z-scheme heterostructures for photocatalytic overall water splitting. *Nature Energy* **6**, 388-397 (2021).
6. Ji S, *et al.* Rare-Earth Single Erbium Atoms for Enhanced Photocatalytic CO₂ Reduction. *Angew Chem Int Ed Engl* **59**, 10651-10657 (2020).
7. Teng Z, *et al.* Atomically dispersed antimony on carbon nitride for the artificial photosynthesis of hydrogen peroxide. *Nature Catalysis* **4**, 374-384 (2021).
8. An X, *et al.* Facilitating Molecular Activation and Proton Feeding by Dual Active Sites on Polymeric Carbon Nitride for Efficient CO₂ Photoreduction. *Angew Chem Int Ed Engl* **61**, e202212706 (2022).
9. Jing L, Zhu R, Phillips DL, Yu JC. Effective Prevention of Charge Trapping in Graphitic Carbon Nitride with Nanosized Red Phosphorus Modification for Superior Photo(electro)catalysis. *Advanced Functional Materials* **27**, (2017).
10. Wang W, *et al.* Femtosecond time-resolved spectroscopic observation of long-lived charge separation in bimetallic sulfide/g-C₃N₄ for boosting photocatalytic H₂ evolution. *Applied Catalysis B: Environmental* **282**, (2021).
11. Zhang X, *et al.* Unraveling the dual defect sites in graphite carbon nitride for ultra-high photocatalytic H₂O₂ evolution. *Energy & Environmental Science* **15**, 830-842 (2022).

REVIEWER COMMENTS

Reviewer #1 (Remarks to the Author):

I have no further questions.

Reviewer #2 (Remarks to the Author):

Well all the comments well addressed, the revised version is acceptable.

Reviewer #3 (Remarks to the Author):

The authors have clarified the synthesis steps in more detail.

They have also included the additional aberration-corrected HAADF-STEM image to demonstrate the existence of single atom Ni atoms. However, the corresponding analyses at Sites 1 and 2 are really not convincing since clustered Ni atoms could be more stabilized than single N atoms. Indeed, the demonstration that single atoms are stable on G-C3N4 should be clearly justified. In this respect, the authors should provide the additional calculation data that the single Ni atoms are more stable than the clustered Ni atoms on G-C3N4. For example, the authors need to show the Gibbs energies for two single Ni atoms and one Ni-Ni cluster adsorbed on G-C3N at room temperature. When the authors should report these data, I recommend including the Gibbs energies at a high-level density functional theory with high-accuracy basis sets. Indeed, without the supporting data relating to these Gibbs energies, it is not recommended for publication of this work to the journal.

Additionally, EXAFS (Figure S25) spectra are different after only 8 cycles of reaction. The authors need to clarify why Ni atoms have different configurations after this recycle test.

Reply letter for manuscript: Revealing structure evolution mechanism of Ni single-atom sites in carbon nitride for efficient photocatalytic H₂O₂ production (NCOMMS-23-18755B).

Reviewer #1 (Remarks to the Author):

I have no further questions.

Reply: Thank you very much for your approval and guidance on our manuscript.

Reviewer #2 (Remarks to the Author):

Well all the comments well addressed, the revised version is acceptable.

Reply: Thank you very much for your approval and guidance on our manuscript.

Reviewer #3 (Remarks to the Author):

The authors have clarified the synthesis steps in more detail.

They have also included the additional aberration-corrected HAADF-STEM image to demonstrate the existence of single atom Ni atoms. However, the corresponding analyses at Sites 1 and 2 are really not convincing since clustered Ni atoms could be more stabilized than single N atoms. Indeed, the demonstration that single atoms are stable on G-C₃N₄ should be clearly justified. In this respect, the authors should provide the additional calculation data that the single Ni atoms are more stable than the clustered Ni atoms on G-C₃N₄. For example, the authors need to show the Gibbs energies for two single Ni atoms and one Ni-Ni cluster adsorbed on G-C₃N at room temperature. When the authors should report these data, I recommend including the Gibbs energies at a high-level density functional theory with high-accuracy basis sets. Indeed, without the supporting data relating to these Gibbs energies, it is not recommended for publication of this work to the journal.

Additionally, EXAFS (Figure S25) spectra are different after only 8 cycles of reaction. The authors need to clarify why Ni atoms have different configurations after this recycle test.

Reply: We sincerely thank you again for your high-quality comments and guidance on the manuscript. For these valuable and constructive suggestions, we have carried out relevant theoretical calculations to further improve the manuscript as requested by the reviewer. Meanwhile, we have made corresponding revisions in the manuscript and highlighted the revised parts with a bright yellow background for easy tracking. With these improvements, we believe that the revised manuscript can better meet the high standards of *Nature Communications*. The point-by-point responses are as follows:

Reviewer #3-Comment 1:

The authors have clarified the synthesis steps in more detail.

Reply: Thank you very much for your approval of the synthesis strategy in the manuscript.

Reviewer #3-Comment 2:

They have also included the additional aberration-corrected HAADF-STEM image to demonstrate the existence of single atom Ni atoms. However, the corresponding analyses at Sites 1 and 2 are really not convincing since clustered Ni atoms could be more stabilized than single N atoms. Indeed, the demonstration that single atoms are stable on G-C₃N₄ should be clearly justified. In this respect, the authors should provide the additional calculation data that the single Ni atoms are more stable than the clustered Ni atoms on G-C₃N₄. For example, the authors need to show the Gibbs energies for two single Ni atoms and one Ni-Ni cluster adsorbed on G-C₃N at room temperature. When the authors should report these data, I recommend including the Gibbs energies at a high-level density functional theory with high-accuracy basis sets. Indeed, without the supporting data relating to these Gibbs energies, it is not recommended for publication of this work to the journal.

Reply: Thanks for your constructive comments. First of all, we have performed additional theoretical calculations according to the calculation requirements and calculation example recommended by the reviewer, which further confirm the rationality and stability of Ni single atoms existing in g-C₃N₄ instead of Ni-Ni clusters. Secondly, we further compared the data in the manuscript with the literature, and explained that there are only Ni single atoms in Ni_{SAPs}-PuCN, and there are no Ni-Ni species in any form.

We carefully answer this question through the following three points. ①. According to the Ni single atoms and Ni-Ni clusters that the reviewer is concerned about, we have supplemented the corresponding theoretical calculations accordingly by using Gaussian 16 C.02¹ at M06-2X functional² and the def2-TZVP³ basis set. Harmonic frequencies have been performed at the same level to confirm that the structure corresponds to the minima on the potential energy surfaces, and Gibbs free energy has been calculated at room temperature (298.15 K) and 1 atm. The calculation results (as shown in Figure R1) show that compared with Ni-Ni clusters, the Gibbs energy of the formation of Ni single atoms on g-C₃N₄ is lower, indicating that Ni single atoms are thermodynamically more stable on g-C₃N₄ than Ni-Ni clusters. ②. Meanwhile, we have performed additional climbing image nudged elastic band calculations⁴ about the activation barriers from the configuration of two Ni single atoms to the Ni-Ni cluster configuration to investigate the possibility of Ni-Ni cluster formation. As shown in Figure R2, the process has to overcome a relatively high energy barrier of 2.14 eV, indicating that Ni single atoms will remain within the cavities, and clustering cannot occur when the deposition is slow and the cavities are not saturated with Ni atoms⁵. **The above thermodynamic and kinetic calculation results clearly demonstrate the good stability of Ni single atoms on g-C₃N₄, responsible for the experimental observations, in which Ni atoms exist in the form of single atoms rather than clusters.** ③. Importantly, Ni single atoms on g-C₃N₄ (Ni_{SAPs}-PuCN) were carefully characterized and demonstrated in our manuscript, and Ni-Ni single atom clusters do not exist on Ni_{SAPs}-PuCN. We describe in detail based on the data in the manuscript and combined with the characterization of single-atom/dual-single-atom catalysts in the current literature.

① Theoretical calculations performed on the reviewer's recommendation:

According to the reviewer's example, we created two Ni single atoms and a Ni-Ni cluster on g-C₃N₄, and analyzed the corresponding Gibbs free energy. The calculation results show that the Gibbs free energy of the single Ni single atom on g-C₃N₄ is much lower than that of Ni-Ni clusters on g-C₃N₄, which indicates that Ni single atoms are easier to form and more stable on g-C₃N₄. The theoretical calculation and analysis performed according to the reviewer's calculation requirements are shown in **Figure R1** below, and this result has been added in the revised **Supplementary Figure 14d-f** (the modified content is highlighted in yellow):

Figure R1 (Added **Supplementary Figure 14d-f** in the revised **Supplementary Information**). The Optimized structures and corresponding Gibbs free energies of (d) two Ni single atoms and (e) one Ni-Ni cluster adsorbed on g-C₃N₄ at room temperature, where the gray, blue and light green balls are C, N, and Ni atoms, respectively. (f) The original data and calculation formula of this theoretical calculation: The Gibbs free energy (hartree) for related molecules and the ΔG (eV) for two single Ni atoms and one Ni-Ni cluster adsorbed on g-C₃N₄ at room temperature (298.15 K) and 1 atm.

The Gibbs free energy for two single Ni atoms (named Ni₂-CN) and one Ni-Ni cluster (named Ni-Ni_{cluster}-CN) adsorbed on g-C₃N₄ at room temperature was studied by density functional theory (DFT), where all structures were optimized via Gaussian 16 C.02 at M06-2X functional and the def2-TZVP basis set. Harmonic frequencies were performed at the same level to confirm that the structure corresponds to the minima on the potential energy surfaces, and

gained Gibbs free energy at room temperature and 1 atm. The optimized structures of Ni₂-CN and Ni-Ni_{cluster}-CN and related Gibbs free energy at room temperature (298.15 K) and 1 atm are shown in **Supplementary Figure 14d-e**. It can be seen that the ΔG (eV) for single Ni atoms adsorbed on g-C₃N₄ is **-2.609 eV**, indicating that the Ni single atoms are preferred to be anchored on g-C₃N₄. When two Ni atoms (Ni-Ni cluster) are adsorbed on g-C₃N₄, ΔG becomes **-1.579 eV (Supplementary Figure 14f)**, indicating that the second Ni atom destabilizes the whole system, which may be that the cavity cannot accommodate two Ni atoms and result in large steric effect. **Considering that lower energy corresponds to greater stability, it is evident that the Ni single atom adsorbed on g-C₃N₄ represents the most reasonable and stable structure.**

② In addition, we further calculated the activation energy barrier from the **two Ni single atoms configuration in g-C₃N₄ to the Ni-Ni cluster configuration to study the possibility of Ni-Ni cluster formation**. The theoretical calculation and analysis are shown in **Figure R2** below, and this result has been added in the revised **Supplementary Figure 14g** (the modified content is highlighted in yellow):

Figure R2 (Supplementary Figure 14g in the revised Supplementary Information). Reaction path from the configuration of **two Ni single atoms to the Ni-Ni cluster configuration on g-C₃N₄**. The energy of initial state is set to be zero. Insets in turn show the configurations of initial state (IS), transition state (TS) and final state (FS).

Next, as shown in **Supplementary Figure 14g**, we calculated the transition state structure and the activation barriers from the configuration of two Ni single atoms to the Ni-Ni cluster configuration on g-C₃N₄. The results (**Supplementary Figure 14g**) show that this process needs to overcome a large energy barrier (2.14 eV), further indicating that it is difficult for isolated Ni single atoms on g-C₃N₄ to form Ni-Ni clusters. **This once again illustrates the**

rationality and stability of Ni single atoms on g-C₃N₄, and is also consistent with the spherical aberration TEM characterization and XAFS results in our manuscript.

Correspondingly, **in the revised manuscript**, the corresponding changes are as follows (the modified content is highlighted in yellow):

1. In **Fig. 2e**, numerous isolated bright spots clearly differ from the contrast of the g-C₃N₄ substrate (See **Supplementary Fig. 14a-c** for more details), indicating that the high density of Ni single atoms was successfully dispersed on the Ni_{SAPs}-PuCN without any nanoparticles or clusters being found.
2. In addition, thermodynamic and kinetic calculation results (**Supplementary Fig. 14d-g**) clearly demonstrate the rationality and stability of Ni single atoms on g-C₃N₄, consistent with experimental characterization results, in which Ni atoms exist as isolated single atoms rather than in the form of clusters.

③. We respect and value the M-M site (Ni-Ni) mentioned by this reviewer very much, so we carefully compare and explain the M-M site reported in the current literature and the single M-site catalyst in the manuscript.

From another point of view, if aggregated metal (M) single atoms (as mentioned by the reviewers: clustered Ni single atoms or Ni-Ni clusters) appear in the sample, M-M coordination peaks will appear in the XAFS characterization of the sample. We illustrate in detail by listing recent articles on clustered M-M dual-single-atom catalysts published in high-quality journals, and comparing the single-atom characterization data in our manuscript, as shown in **Figure R 3** below:

Figure R3. The comparison between the core data of M-M double-single-atom catalysts published in high-quality journals reported in the literature and our work. (a-c) Schematic diagram of the dual Co-Co single-atom photocatalyst⁶, spherical aberration electron microscope characterization Fourier transformation of the EXAFS spectra. (d-f) Schematic diagram of the dual Pt-Fe single-atom electrocatalyst⁷, spherical aberration electron microscope characterization Fourier transformation of the EXAFS spectra. (g-i) Schematic diagram of the dual Ni-Ni single-atom catalyst⁸, spherical aberration electron microscope characterization Fourier transformation of the EXAFS spectra. (j-l) Schematic diagram of the Ni single atoms photocatalyst (our work in this manuscript) spherical aberration electron microscope characterization Fourier transformation of the EXAFS spectra.

(1) As shown in **Figure R3a**, Tae Kyu Kim's group synthesized Co-Co dual-single-atom catalysts on g-C₃N₄. The spherical aberration electron microscopy showed that a large number of clustered Co-Co dual-single-atom pairs appeared in the catalyst (**Figure R3b**),

and the EXAFS spectra confirmed that there was a strong Co-Co coordination (see the position of the yellow arrow in Figure R3c) in the catalyst⁶.

- (2) As shown in Figure R3d, Liu et al. reported Pt-Fe dual single-atom electrocatalysts. The spherical aberration electron microscopy showed that many clustered Pt-Fe dual-single-atom pairs appeared in the catalyst (Figure R3e), and EXAFS spectra also proved the Pt-Fe bond in the dual-single-atom (see the position of the yellow arrow in Figure R3f)⁷.
- (3) Likewise, as shown in Figure R3g, Zhang et al. designed a Ni-Ni dual-single-atom electrocatalyst (Ni₂NC). Most of the clustered Ni-Ni dual-single-atoms in the Ni₂NC were characterized by spherical aberration electron microscopy (Figure R3h). Importantly, EXAFS spectroscopy sensitively detects the Ni-Ni bonds formed between dual-single-atom in Ni₂NC, and contrasts the difference between Ni single atoms catalyst (Ni₁NC) without Ni-Ni bonds (see the position of the yellow arrow in Figure R3i)⁸.
- (4) For comparison, in Figure R3j–l, we synthesized Ni single-atom catalyst on g-C₃N₄, which was mutually verified by multiple characterization means in the manuscript. The spherical aberration electron microscope characterization (Figure R3k) shows that Ni single atoms are highly disorderly dispersed on g-C₃N₄, and the EXAFS spectra (Figure R3l) suggests that Ni single atoms only coordinate with N atoms, and there is no Ni-Ni bond between Ni single atoms (see the Ni_{SAPs}-PuCN sample corresponding to the position of the red arrow in Figure R3l). Meanwhile, it should be noted that the distance between two isolated Ni single atoms in site 2 (Supplementary Figure 14c) is too far (~ 0.8 nm) to form the Ni-Ni bond.

The above detailed comparison further illustrates the absence of any form Ni-Ni species in Ni_{SAPs}-PuCN in our manuscript. For the synthesis of dual-single-atom catalysts with strong metal-metal (M-M) interactions, ingenious strategies such as designing ligands, selection of single-atom salts, and regulating substrate structures are often required. However, the synthetic strategy proposed in our manuscript is only applicable to highly loaded single-atom catalysts.

In summary, through the above supplementary theoretical calculations, detailed experimental data explanations, and literature comparisons, the rationality and stability of the existence of only Ni single atoms in Ni_{SAPs}-PuCN in our manuscript is fully confirmed. Thanks again to this reviewer for raising these critical issues.

Reviewer #3-Comment 3:

Additionally, EXAFS (Figure S25) spectra are different after only 8 cycles of reaction. The authors need to clarify why Ni atoms have different configurations after this recycle test.

Reply: Thanks again for your careful review and kind reminder. ①. According to the EXAFS spectra (Supplementary Figure 25d) in the manuscript, there is no change in the Ni single-atom coordination structure and configuration in the Ni_{SAPs}-PuCN after reaction cycles. The

small bulge (at around 1 Å, does not correspond to any metal-coordinating atom distance) in the EXAFS spectra of Ni_{SAPs}-PuCN after 8 cycles of reaction comes from the background noise during the test (common experimental error), which has no effect on the core Ni-N coordination peak and the conclusion of the manuscript. ②. This is due to common background noise and experimental error that cause the EXAFS spectra to "look" changed, which may have influenced the reviewers' judgment. We apologize for this. Further, under the guidance of XAFS experts, the experimental error caused by background noise can be reduced by optimizing the adjustment parameters to avoid misleading readers. Adjusted EXAFS spectra were placed in the revised Supplementary Information (Supplementary Figure 25d). ③. Meanwhile, we supplemented the molecular dynamics (MD) calculations to further confirm the thermodynamic stability of the Ni_{SAPs}-PuCN model (added in the revised Supplementary Figure 25g).

Detailed explanation and adjustment are explained from the following two points:

①. According to the working principle of XAFS, the EXAFS spectral data (Supplementary Figure 25d) of the samples before and after the reaction cannot be perfectly overlapping, and there are also factors such as common background interference and unavoidable test errors. It should be noted that the position and intensity of the core Ni-N coordination peaks in the EXAFS spectra (**Supplementary Figure 25d**) of the Ni_{SAPs}-PuCN before and after the reaction are almost unchanged, which strongly illustrates the stability of the Ni single-atom configuration in the samples after the reaction.

Next, we combine the principle of XAFS with **Supplementary Figure 25c-d** for a detailed description, as shown in **Figure R4** below:

Figure R4 (Supplementary Figure 25c-d) Structural stability of Ni_{SAPs}-PuCN after photocatalytic 8-cycle reaction. (c) Ni K-edge XANES spectra of the Ni_{SAPs}-PuCN sample before and after the 8 cycles of reaction. (d) Fourier transform of EXAFS spectra of Ni_{SAPs}-PuCN before and after 8 cycles of reaction (For explanatory purposes, we have only added additional markers to **Supplementary Figure 25d** in this reply letter).

Firstly, X-ray absorption fine structure (XAFS) is composed of X-ray absorption near-edge structure (XANES) and extend X-ray absorption fine structure (EXAFS). The EXAFS oscillator function $\chi(k)$ can be expressed as:

$$\chi(k) = \sum_j \frac{N_j S_0^2 f_j(k)}{k} \int_0^\infty \frac{g(R_j)}{R_j^2} e^{-2r_j/\lambda(k)} \sin[(2kR_j + \delta(k) + 2\phi_c(k)] dR$$

Where N_j represents the number of coordination atoms, and S_0^2 is the amplitude attenuation factor. R is the distance between the absorbing atom and the scattering atom, and $g(R_j)$ is the pair distribution function of the scattering atom. $\lambda(k)$ is the mean free path of the excited photoelectron. $f_j(k)$ and $\delta(k)$ are the backscattering amplitude and the scattering phase shift function of the scattering atom, respectively, and $\phi(k)$ is the phase shift function of the absorbing atom. In a word, the EXAFS is mainly related to the coordination number, bond length, coordination atom and disorder of the central atom. **However, the experimental light source, the experimental error and the noise of the detector will do have the weak influence on the test results (Sci China Mater 2015, 58: 313–341).** Meanwhile, EXAFS spectra are obtained by measuring the photoelectron emission of X-rays after they have been absorbed in the material. When interpreting EXAFS data for single-atom catalysts, one usually focuses on the range of distances between 1.5 Å and 3.0 Å, which is the effective working range of the EXAFS

technique.

Furthermore, the very weak “bulge” (approximately close to 1 Å, marked with a red dotted line) in the **Figure R4d (Supplementary Figure 25d)** represent the background noise due to effects of the low metal content and multiple scattering, and these do not represent the change of coordination structure for the Ni sites. As common knowledge, the coordination peaks in EXAFS appear in the range of 1.5 Å to 3 Å, corresponding to the interaction between the metal and its coordination atoms. In **Figure R4d (Supplementary Figure 25d)**, the position of the “small bulge” appears close to 1 Å, which does not fit the metal-coordinating atom distance.

Importantly, as seen in the **Supplementary Figure 25d**, the dominant peak at ~1.69 Å assigned to Ni-N coordination shows consistent peak intensity and peak position, clearly demonstrate the similar coordination structures of Ni site before and after the reaction, which is because peak strength and position are related to coordination number and coordination bond length, respectively. Above all, the Ni_{SAPs}-PuCN exhibits a stable structure without any change in configurations before and after 8 cycles of reaction.

②. Supplementary Figure 25d after adjustment.

We attach great importance to the issues raised by the reviewer, and to avoid confusing readers, we adjusted the parameters to reduce the background noise interference of the EXAFS spectra in Ni_{SAPs}-PuCN after the reaction, under the guidance of XAFS experts. The specific operation is as follows: Using the ATHENA module implemented in the IFEFFIT software packages (*J. Synchrot. Radiat.* 8, 322–324, 2001). Subsequently, k^2 -weighted $\chi(k)$ data ranging from 2.3-10.1 Å⁻¹ were Fourier transformed to real space using a Hanning windows ($dk = 1.0$ Å⁻¹) to separate the EXAFS contributions from different coordination shells. In order to realize the best background removal for EXAFS data analysis, we selected the larger R_{bkg} values of 1.1 Å to remove low frequency noise, and the spline range of 0-650 eV was unified to reduce background noise. The adjusted EXAFS spectra were added in the revised **Supplementary Figure 25**, as shown in **Figure R5** below (the modified content is highlighted in yellow):

Figure R5 (Supplementary Figure 25c-d in the revised Supplementary Information). (c) Ni K-edge XANES spectra of the Ni_{SAPs}-PuCN sample before and after the 8 cycles of reaction. (d) Fourier transform of EXAFS spectra of Ni_{SAPs}-PuCN before and after 8 cycles of reaction.

As shown in **Supplementary Figure 25c**, the Ni K-edge X-ray absorption near-edge structure (XANES) spectra of Ni_{SAPs}-PuCN before and after the reaction were basically changed little, proving that the valence state of Ni in Ni_{SAPs}-PuCN was basically unchanged before and after the reaction. Meanwhile, **Supplementary Figure 25d** shows the Fourier transform of the extended X-ray absorption fine structure (FT-EXAFS) spectra of Ni_{SAPs}-PuCN before and after the reaction. It can be seen in **Supplementary Figure 25d** that the Ni_{SAPs}-PuCN after 8 cycles reaction also exhibits a main peak around 1.69 Å (Ni-N). Meanwhile, for the core Ni-N coordination peak, the Ni_{SAPs}-PuCN before and after the cyclic reaction has the same peak intensity and peak position, which indicated that the Ni single-atom coordination structure in Ni_{SAPs}-PuCN did not change after the cyclic reaction. Furthermore, Ni_{SAPs}-PuCN after the cyclic reaction did not detect the appearance of Ni-Ni bond (**Supplementary Figure 25d**), indicating that there was no agglomeration of Ni single atoms.

③. To further investigate the stability of Ni single-atom coordination in Ni_{SAPs}-PuCN, we confirmed the thermodynamic stability of the Ni_{SAPs}-PuCN model through molecular dynamics (MD) calculations (the calculation results are consistent with the requirements for the stability of single-atom catalysts reported in the literature⁹). The calculated results and analysis are shown in **Figure R6** below (and this part was added in the revised **Supplementary Figure 25g**, the modified content is highlighted in yellow):

Figure R6 (Supplementary Figure 25g in the revised Supplementary Information). DFT total energy versus total simulation time at 500 K for the Ni_{SAPs}-PuCN structural model (Ni-N₃). The inset shows the structure at 10 ps.

In addition, in Supplementary Figure 25g, the molecular dynamics (MD) calculations performed on the Ni_{SAPs}-PuCN model at a temperature of 500 K are shown (total 10 ps). The fluctuation of the total energy of the Ni_{SAPs}-PuCN model with time evolution is shown in Supplementary Figure 25g. It is noteworthy that the energy consistently oscillates in proximity to the equilibrium state throughout the entire simulation, without any observed Ni single-atom coordination change or structural damage, indicating the thermodynamic stability of this Ni-N₃ model.

Correspondingly, in the revised manuscript, the corresponding changes are as follows (the modified content is highlighted in yellow):

Meanwhile, Ni_{SAPs}-PuCN exhibited excellent performance cyclability (Supplementary Fig. 24) and structural stability (Supplementary Fig. 25a-f). The thermodynamic stability of the Ni_{SAPs}-PuCN model was also confirmed by molecular dynamics (MD) calculations (Supplementary Fig. 25g).

Finally, once again sincerely thank this reviewer for your valuable time and constructive comments to further improve the research quality of the manuscript.

References

1. Frisch M, *et al.* Uranyl Extraction by N, N-Dialkylamide Ligands Studied by Static and Dynamic DFT Simulations. In: *Gaussian 09*. Gaussian Inc Wallingford (2009).
2. Zhao Y, Truhlar DGJTca. The M06 suite of density functionals for main group thermochemistry, thermochemical kinetics, noncovalent interactions, excited states, and transition elements: two new functionals and systematic testing of four M06-class functionals and 12 other functionals. **120**, 215-241 (2008).
3. Weigend F, Ahlrichs RJPCCP. Balanced basis sets of split valence, triple zeta valence and quadruple zeta valence quality for H to Rn: Design and assessment of accuracy. **7**, 3297-3305 (2005).

4. Henkelman G, Uberuaga BP, Jónsson HJTJocp. A climbing image nudged elastic band method for finding saddle points and minimum energy paths. **113**, 9901-9904 (2000).
5. Vilé G, *et al.* A Stable Single-Site Palladium Catalyst for Hydrogenations. *Angewandte Chemie International Edition* **54**, 11265-11269 (2015).
6. Wang J, *et al.* Highly Durable and Fully Dispersed Cobalt Diatomic Site Catalysts for CO₂ Photoreduction to CH₄. *Angew Chem Int Ed Engl* **61**, e202113044 (2022).
7. Zhou W, *et al.* Regulating the scaling relationship for high catalytic kinetics and selectivity of the oxygen reduction reaction. *Nat Commun* **13**, 6414 (2022).
8. Hao Q, *et al.* Nickel dual-atom sites for electrochemical carbon dioxide reduction. **1**, 719-728 (2022).
9. Yang T, *et al.* Coordination tailoring of Cu single sites on C₃N₄ realizes selective CO₂ hydrogenation at low temperature. *Nat Commun* **12**, 6022 (2021).

REVIEWERS' COMMENTS

Reviewer #3 (Remarks to the Author):

The revised manuscript is well prepared, so considered to be ready for publication.